# Predicting cell-to-cell communication networks using NATMI

Rui Hou [1], Elena Denisenko[1], Huan Ting Ong [2], Jordan A. Ramilowski [3,4] & Alistair R. R. Forrest [1,4✉]

Development of high throughput single-cell sequencing technologies has made it cost-effective to profile thousands of cells from diverse samples containing multiple cell types. To study how these different cell types work together, here we develop NATMI (Network Analysis Toolkit for Multicellular Interactions). NATMI uses connectomeDB2020 (a database of 2293 manually curated ligand-receptor pairs with literature support) to predict and visualise cell-to-cell communication networks from single-cell (or bulk) expression data. Using multiple published single-cell datasets we demonstrate how NATMI can be used to identify (i) the cell-type pairs that are communicating the most (or most specifically) within a network, (ii) the most active (or specific) ligand-receptor pairs active within a network, (iii) putative highly-communicating cellular communities and (iv) differences in intercellular communication when profiling given cell types under different conditions. Furthermore, analysis of the Tabula Muris (organism-wide) atlas confirms our previous prediction that autocrine signalling is a major feature of cell-to-cell communication networks, while also revealing that hundreds of ligands and their cognate receptors are co-expressed in individual cells suggesting a substantial potential for self-signalling.

[1] Harry Perkins Institute of Medical Research, QEII Medical Centre and Centre for Medical Research, The University of Western Australia, Nedlands, WA 6009, Australia. [2] Ear Science Institute Australia, and Ear Sciences Centre, The University of Western Australia, Nedlands, WA 6009, Australia. [3] Advanced Medical Research Center, Yokohama City University, Yokohama, Kanagawa 236-0004, Japan. [4] RIKEN Center for Integrative Medical Sciences, Yokohama, Kanagawa 230-0045, Japan. ✉email: alistair.forrest@gmail.com

In 2015, we published the first draft map of predicted cell-to-cell communication between major human cell types based on the expression levels of 708 ligands and 691 receptors measured in 144 purified human primary cell types, and a manually curated set of 2422 human ligand–receptor pairs (1894 pairs had literature support)[1]. Our major findings were that any given cell type expresses tens to hundreds of different ligands and receptors, that extensive autocrine signalling is a common feature of all cell types studied thus far and that cell-to-cell communication networks are highly connected via hundreds of predicted ligand–receptor paths.

Recently, with the availability of high throughput single-cell platforms[2–10], expression profiling across an ever-growing variety of cell types from dissociated tissues without prior purification has become common. To understand how individual cells and cell types within these complex samples communicate, multiple groups have since used our ligand–receptor pairs from 2015[1] to infer cell-to-cell communication in developing heart[11], kidney[12], liver[13], lung[14], cancer[15,16], and cortex[17].

Here, we present NATMI (Network Analysis Toolkit for Multicellular Interactions), a user-friendly tool primarily designed to process single-cell gene expression datasets, which can also be applied to bulk transcriptomics and proteomics data. In brief, NATMI uses connectomeDB2020 (a newly updated curated ligand–receptor pair list) or user-specified ligand–receptor pairs to predict and visualise the network of cell-to-cell communication between cell types (or clusters) in these datasets (Supplementary Fig. 1).

We demonstrate several ways NATMI can be used to summarise and extract biological insights from cell-to-cell communication networks (Supplementary Fig. 1). Specifically, NATMI can: (1) show all cell types predicted to communicate via a user-specified ligand–receptor pair (Supplementary Fig. 1d); (2) show all ligand–receptor pairs used for communication between a user-specified pair of cell types (Supplementary Fig. 1e); (3) summarise the entire communication network to show how strongly or specifically each cell type in a complex sample communicates to every other cell type, thus identifying highly communicating cell pairs or communities (Supplementary Fig. 1f); and (4) compare communication networks from two different conditions and identify edges (ligand–receptor pairs) that differ (delta network) between them (Supplementary Fig. 1g). NATMI (Python script) and the updated ligand–receptor lists are freely available at https://github.com/forrest-lab/NATMI/.

## Results

**Updated ligand–receptor pair lists**. To facilitate the exploration of intercellular interactions, in 2015 we published a set of 1894 ligand–receptor pairs with primary literature support and an additional 528 putative pairs (secreted and plasma-membrane proteins with high throughput protein–protein interaction (PPI) evidence)[1]. Here, we present connectomeDB2020, an updated set of 2293 ligand–receptor pairs with primary literature support (Supplementary Data 1). These consist of 1751 pairs from our 2015 resource, 121 pairs from CellphoneDB v2.0[18], 50 pairs from RNA-Magnet[19], 22 pairs from SingleCellSignalR[20], 9 pairs from ICELLNET[21], and 340 new manually curated pairs predominantly reported since the original publication. Supplementary Data 1 also lists pairs from each resource excluded due to lack of primary evidence (these include 143 of our original pairs removed after user feedback and further checks revealed that the PubMed ID supplied from HPRD[22] was incorrect and did not support the interaction). To allow users to discriminate contact-dependent signalling from soluble ligand-mediated signalling, we have now annotated all ligands as either secreted,

plasma membrane or both (Methods). Additionally, to facilitate the use of NATMI for other organisms, we provide inferred ligand–receptor pairs for multiple vertebrate species based on human ortholog mappings provided in the NCBI HomoloGene Database[23].

**Predicting cell-to-cell communication using NATMI**. The workflow for cell-to-cell communication analysis in NATMI starts by the user providing input data files of gene expression and cell labels for single-cell data analysis. Using the connectomeDB2020 ligand–receptor pair list described above, NATMI extracts the expression levels of every ligand and receptor expressed in each cell type (Supplementary Fig. 1a–c). Edges between any pair of cell types are then predicted based on the expression of the ligand in one cell type and the expression of its cognate receptor in the other cell type (Supplementary Fig. 2a). As cells can express multiple ligands and cognate receptors, cell pairs are connected by multiple edges that are accordingly weighted by the expression of ligands and receptors in these cell types (Supplementary Fig. 2b). Moreover, the same ligands and receptors can be expressed by multiple cell types. This makes any given ligand–receptor pair a hyperedge that can connect multiple cell types and the underlying structure of a real cell-to-cell communication network—a weighted-directed-multi-hyperedge network (Supplementary Fig. 2c). In NATMI, however, we reduce these to weighted-directed-multi-edge networks (Supplementary Fig. 2d) and for simplicity refer to these as 'cell-to-cell communication networks'. Lastly, NATMI introduces the concept of cell-connectivity-summary networks that merge the many ligand–receptor edges drawn from one cell type to another into a combined weighted cell-connectivity-summary edge to summarise how strongly (or specifically) each cell type is communicating to another cell type (Supplementary Fig. 1f).

**Edge weights in cell-to-cell communication networks**. For each analysed dataset NATMI creates an edge file that summarises the levels and fractions of cells in each cell type expressing each ligand and receptor. From this it calculates two different edge weights. The mean-expression edge weights are calculated by multiplying the mean-expression level of the ligand in the sending cell type by the mean expression of the receptor in the target cell type. This weighting is useful to emphasise highly expressed ligands and receptors but provides no discrimination between cell-type-specific and housekeeping edges. The specificity-based edge weights, on the other hand, help identify the most specific edges in the network regardless of expression levels and are calculated as the product of the ligand and receptor specificities, where each specificity is defined as the mean expression of the ligand/receptor in a given cell type divided by the sum of the mean expression of that ligand/receptor across all cell types. The specificity-based edge weights range from 0 to 1 where a weight of 1 means both the ligand and receptor are only expressed in one (not necessarily the same) cell type.

To better demonstrate the utility of different edge weighting metrics in NATMI, we reanalysed communication between 12 defined cell types in a previously published cardiac dataset[11] and predicted 126,738 cell-ligand–receptor-cell edges (Supplementary Data 2). Ranking edges based on expression weighting (Fig. 1a, shows the edges for the top 20 most highly expressed ligand–receptor pairs) differed substantially from those based on specificity weighting (Fig. 1b, shows the edges for the top 20 most specific ligand–receptor pairs). Notably, the ligand–receptor pairs ranked by expression weighting were also detected in a broader range of communicating cell type pairs (noticeable when

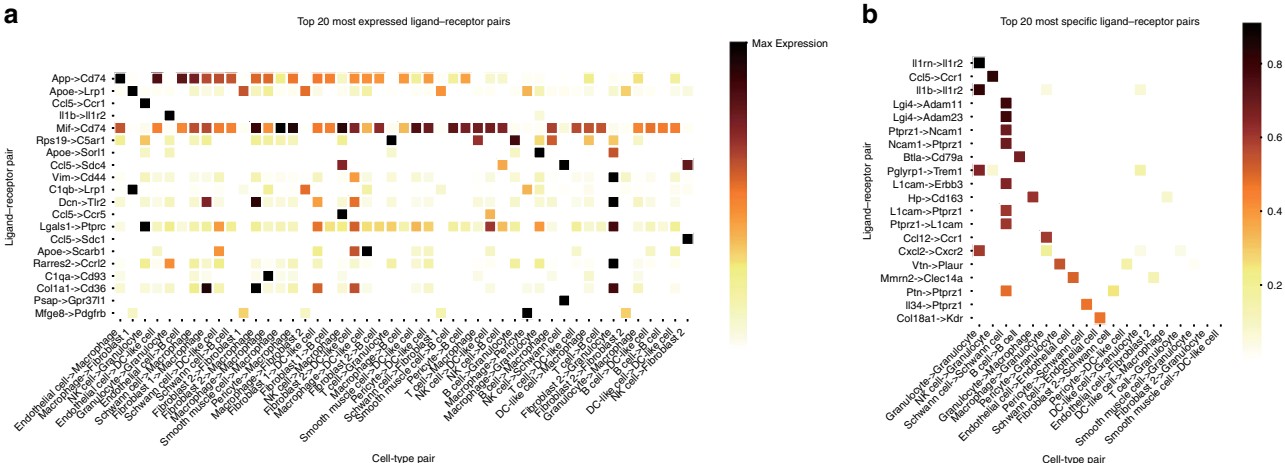

**Fig. 1 Top 20 ligand–receptor pairs ranked by expression or specificity weighting in the Skelly et al.[11] dataset.** Ligand-receptor pairs ranked by **a** expression (product of mean ligand expression level × mean receptor expression level). Rows are scaled by max. **b** Specificity (product of ligand specificity × receptor specificity). Ligand–receptor pairs with high expression weights (**a**) may be broadly used by many cell-type pairs, while those with high specificity weights (**b**) tend to be limited to one or only a few cell-type pairs.

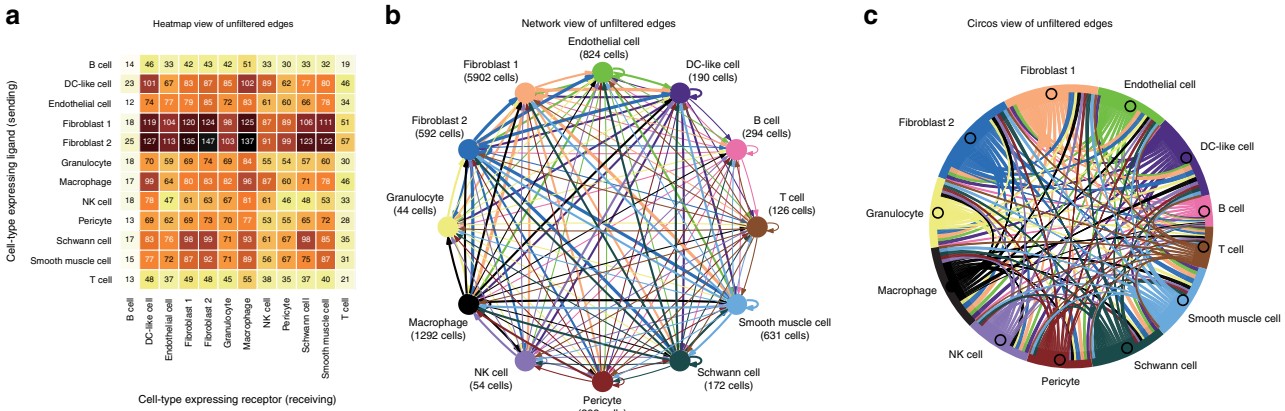

**Fig. 2 Cell-connectivity-summary-network visualisations in NATMI.** To identify cell types that are communicating 'more' than others in Skelly et al.[11], the number of ligand–receptor pairs (detected in more than 20% of cells) connecting each pair of cell types were counted. Resulting simple edge-count-based cell-connectivity-summary network is shown using three distinct visualisation methods available in NATMI. **a** Heatmap view. Rows indicate cells expressing the ligands and columns indicate cells expressing the receptors. Number of ligand–receptor pairs connecting the cell–cell pairs are indicated and coloured relative to max count. **b** Network-graph view and **c** Circos-view for the same network. For **b**, **c**, edges are coloured by ligand expressing cell type, arrows indicate target cell type and thickness is proportional to the number of ligand–receptor pairs connecting the two cell types. The NATMI directed heatmap visualisation is asymmetric. This recognises that cell communication is directional and that although one cell type may send many signals to another there is no guarantee that it is reciprocated.

comparing the larger width of Fig. 1a to that of Fig. 1b) and thus kept potentially less informative housekeeping edges.

**Cell-connectivity-summary networks.** One of the primary aims of cell-to-cell communication network analysis is to identify which cell types are mutually coordinating their activities by ligand–receptor-mediated communication. Our analyses indicate, however, that all cell types have substantial potential to communicate with each other. Consequently, this leads to the question of which cell types are communicating the most? Or the most specifically? The simplest strategy to measure the degree of communication from one cell type to another is to count the number of ligand–receptor pairs connecting them. Figure 2 summarises the cell-connectivity-summary network for the above cardiac dataset based on simple edge count using three different visualisations—heatmap, network graph, and circos plot. High connectivity of these networks makes the network graph and circos views not easily interpretable due to over-plotting issues.

The heatmap view, however, avoids the problem and reconfirms one of the major predictions of the cardiac study[11] that fibroblasts are the most trophic, with edges from fibroblasts to 10 of the 12 cell types dominating the network. In Fig. 3, we show how NATMI can be used to filter the network based on expression levels or specificities of the ligands and receptors involved.

Filtering by expression weights (Fig. 3a) can provide users a higher confidence that the ligands and receptors are expressed at sufficient levels. For the cardiac dataset, we explored both the filtered by expression and unfiltered network (Fig. 2) yielded, however, a similar conclusion that the fibroblasts are the most trophic. In contrast, filtering on specificity weights (Fig. 3b) highlights a different set of top cell-to-cell pairs. In particular, autocrine signalling of Schwann cells, endothelial cells and granulocytes, fibroblast and Schwann cell signalling to endothelial cells, and fibroblast, granulocyte and pericyte signalling to granulocytes is highlighted while the broad signalling from fibroblasts seen in the unfiltered and expression filtered networks

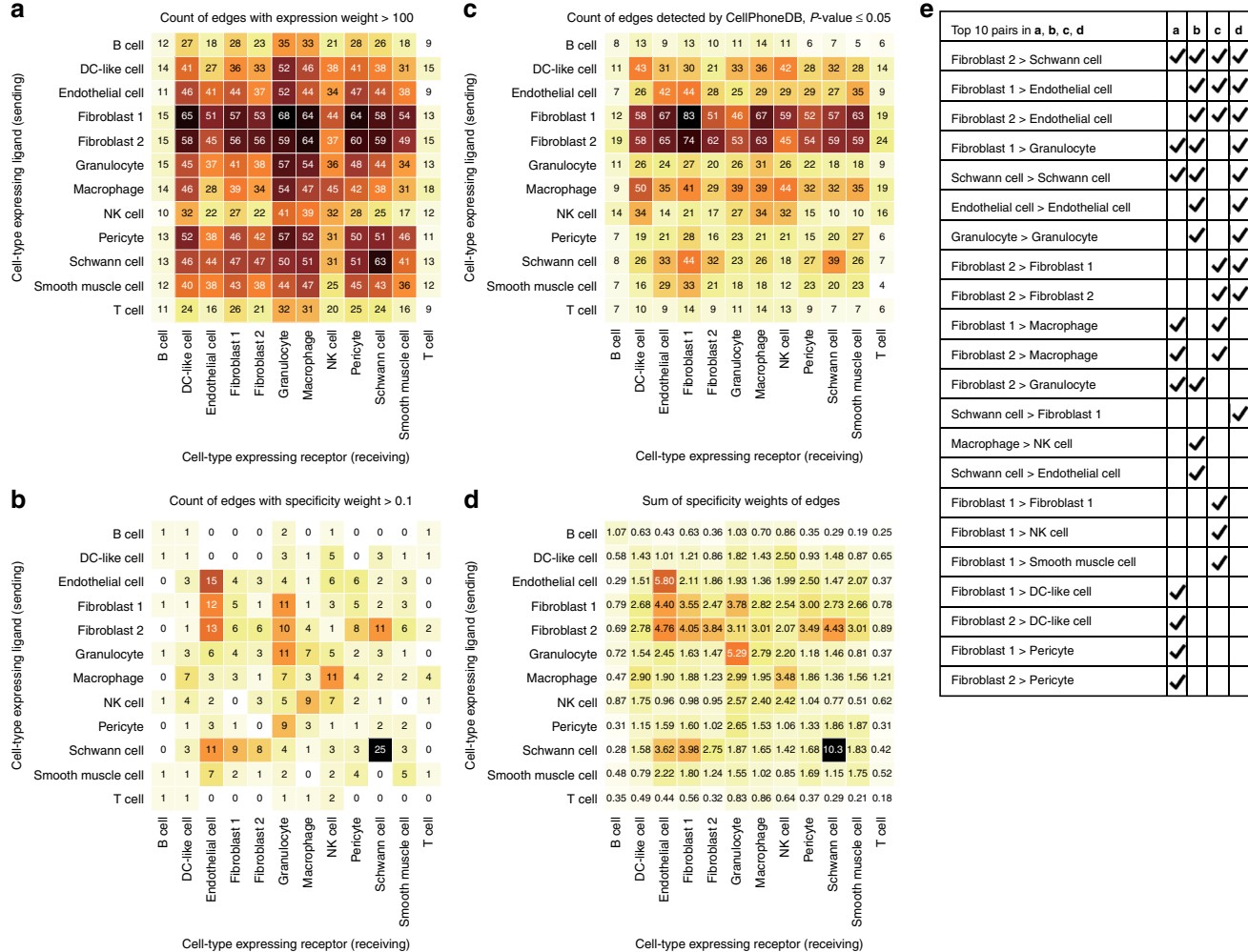

**Fig. 3 Impact of different edge filters and metrics on identifying top communicating cell-type pairs.** Heatmap views of the Skelly et al.[11] cell-connectivity-summary network shown in Fig. 2 after filtering ligand–receptor pairs based on **a** mean-expression weight (≥100), **b** specificity (≥0.1) or **c** $p$ values obtained by using CellPhoneDB[18] (≤0.05). As an alternative to hard filtering the network, view in **d** is weighted by the sum of the specificities. **e** Compares the top 10 communicating cell type pairs identified in **a–d**.

is diminished. We next compared our results with those obtained by filtering edges based on $p$ values calculated by CellPhoneDB[18]. The resulting heatmap (Fig. 3c) is similar to that observed for the expression filtered network (Fig. 3a) suggesting NATMI may better highlight high specificity edges. (Note, the heatmap shown in Fig. 3c should not be confused with those generated by CellPhoneDB which are symmetric. NATMI heatmaps are asymmetric and have direction from the ligand expressing cell type to the receptor expression cell type.) Lastly, the network can also be summarised using the summed-specificity weights between each cell type pair (Fig. 3d). This generates a similar network to that in Fig. 3b, without requiring to set an arbitrary threshold on specificity. Noticeably, as each approach generates a different view of the network and highlights different most-communicating cell type pairs (Fig. 3e), users need to consider these differences when interpreting their own cell-to-cell communication networks. In NATMI, the user can choose any of its built-in approaches, however, we recommend to use summed specificity for most analyses as this captures specific signalling between cell types (Fig. 3d). Different edge filtering methods are further explained in a concept Supplementary Fig. 3.

**Application of NATMI to an organism-wide single-cell dataset.** One of the ultimate aims of developing intercellular

communication network methods is to understand the general principles of cell-to-cell communication within multicellular organisms. Previously, analysis of the FANTOM5 (bulk expression) dataset[1] revealed that most cell types express tens to over a hundred different ligands and receptors, and that hematopoietic cells tend to express fewer ligands and receptors than cells from other lineages. Importantly, it also predicted a substantial potential for autocrine signalling, with over 50% of the ligands and receptors detected in each cell type having cognate partners expressed in the same cell type. To examine whether these observations were consistent when using single-cell expression data, we repeated the analysis by applying NATMI to the Tabula Muris atlas[24] (a mouse cell atlas containing 44,949 FACS sorted cells from 20 organs and classified into 117 organ-resident cell types).

**Autocrine, self, and intra-organ signalling in Tabula Muris.** At a detection rate threshold of 20% (commonly applied to single-cell datasets[11,25]), most cell types in the Tabula Muris dataset expressed over a hundred ligands and receptors, with hematopoietic cell types expressing fewer ligands/receptors than other lineages (Supplementary Fig. 4a). Notably, almost half of the ligands detected in any given cell type in Tabula Muris had cognate receptors (and vice versa) detected in the same cell type

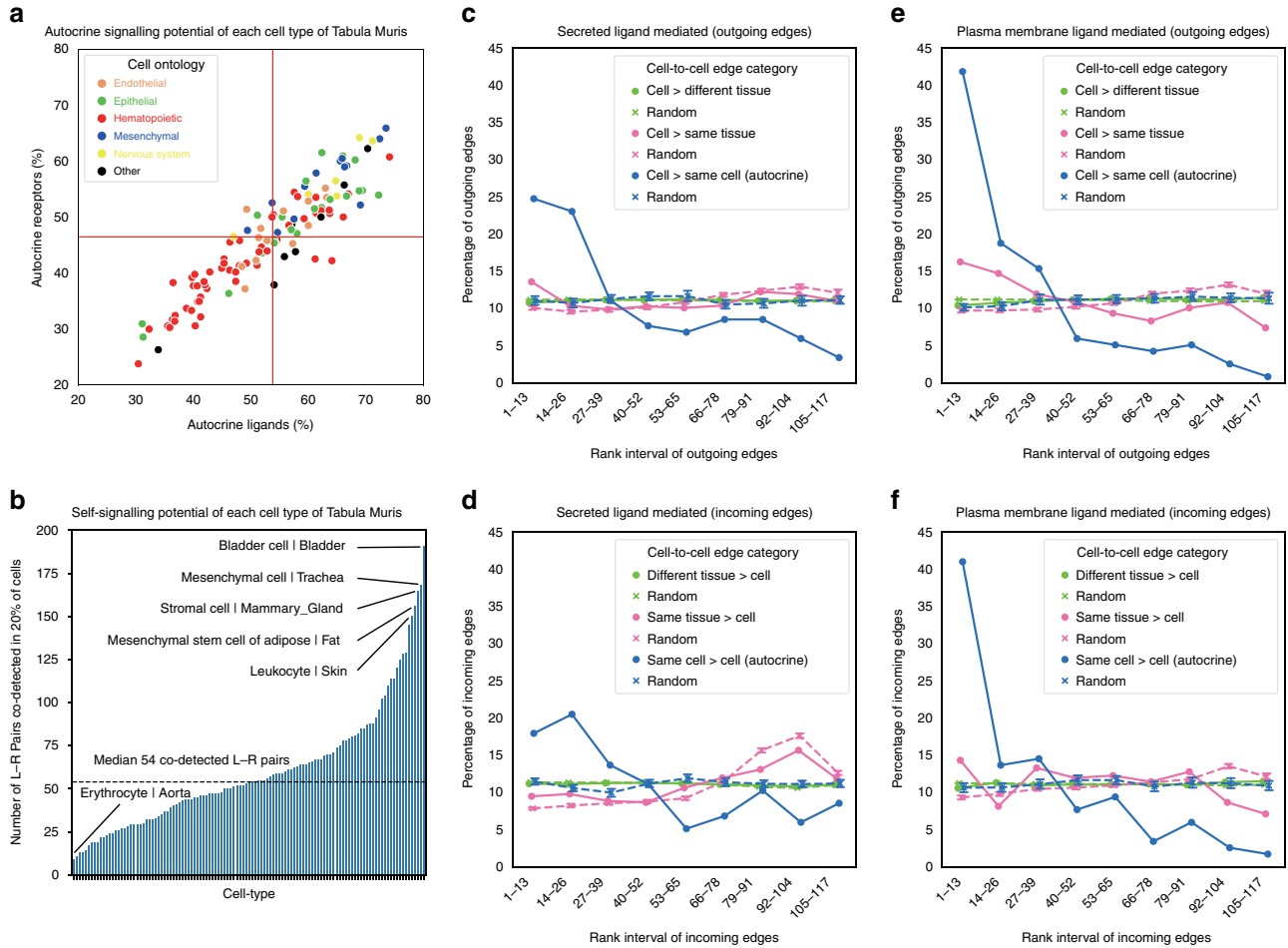

**Fig. 4 Autocrine and self-signalling potential in Tabula Muris.** NATMI was run across the 117 cell types identified in the Tabula Muris dataset (at the detection rate threshold of 20%). **a** Autocrine signalling potential of each cell type in Tabula Muris. Each point corresponds to a cell type, and colours indicate broad lineage classes. X-axis shows the fraction of ligands expressed by a given cell type where the cognate receptor is also expressed in the same cell type. Y-axis shows the reciprocal for the fraction of receptors on a given cell type where the cognate ligand is also expressed in the same cell type. Red lines show the mean fractions of ligands and receptors. **b** Self-signalling potential of each cell type in Tabula Muris. The dashed line indicates the median number of ligand–receptor pairs co-detected in more than 20% of cells of each cell type (at 10CPM threshold). In **c–f**, cell-to-cell pairs are classified as autocrine (blue), intra-organ (pink), and inter-organ (green) summary edges. The summary edges are then ranked by summed specificity and the distribution of ranks for each of the three classes shown. Error bars represent standard deviation of ranks. **c** The distribution of ranks of secreted ligand mediated outgoing edges. **d** The distribution of ranks of secreted ligand mediated incoming edges. **e** The distribution of ranks of plasma-membrane ligand mediated outgoing edges. **f** The distribution of ranks of plasma-membrane ligand mediated incoming edges. The average distributions of autocrine, intra-organ and inter-organ edges through 100× randomised ligand–receptor pairs are shown as dashed lines.

further confirming our previous prediction of large potential for autocrine signalling in cell-to-cell communication networks (Fig. 4a).

Next, using the single-cell resolution data we could discriminate whether a ligand and its cognate receptor were actually expressed in the same cell or two different cells of the same cell type (a distinct advantage over bulk measurements where autocrine and self-signalling cannot be discriminated). Across the 117 cell types in Tabula Muris, a median of 54 ligand–receptor pairs were co-detected in at least 20% of the same cells (from each cell type) at an expression threshold of 10 counts per million (CPM, Fig. 4b, Supplementary Data 3 lists the ligand–receptor pairs co-detected in each cell type). This extends on our original observations of substantial autocrine signalling potential using the FANTOM5 data[1] and is the first finding that cognate ligands and receptors are co-expressed in a substantial fraction of single cells suggesting their potential for self-signalling. Moreover, we observed that a substantial fraction of all ligand–receptor pairs (31%, 719/2293) were co-detected in at

least 20% of cells of at least one cell type at an expression threshold of 10CPM (Supplementary Data 4 and Supplementary Fig. 4b).

To examine autocrine signalling in more detail, we next generated cell-connectivity-summary networks weighted by the summed specificity between each of the 117 cell types in the Tabula Muris dataset (Supplementary Data 5 and Supplementary Fig. 5). To discriminate between contact-dependent and contact-independent signalling, we generated two separate analyses based on pairs that involved either secreted ligands or plasma-membrane ligands.

For each of these 117 cell types, we ranked their connections (outgoing and incoming edges) by their summed-specificity weights and classified cell-to-cell summary edges as autocrine, intra-organ (excluding autocrine) and inter-organ. In Fig. 4c–f, we then plotted the distribution of ranks for autocrine ('blue'), intra-organ ('pink'), and inter-organ ('green') summary edges. When compared, autocrine edges had higher rankings, meaning that, on average, autocrine edges tend to be more specific than

intra-organ and inter-organ edges (solid lines in Fig. 4c–f), whereas repeated analyses using randomly permuted receptor-ligand pairs abolished these differences (dashed lines in Fig. 4c–f). A slight enrichment was also observed for intra-organ signalling for outgoing plasma-membrane ligand-mediated edges (Fig. 4e), while no such enrichment was found for the secreted ligand-mediated edges (Fig. 4c, d) and for the plasma-membrane receiving edges (Fig. 4f).

To test whether the observed higher autocrine rankings depend on the edge weighting method used, we repeated the analysis using simple edge counts and summed-expression weights and again observed higher (albeit to a lesser degree than shown in Fig. 4) rankings of autocrine signalling for the edge-count-based analysis (Supplementary Fig. 4c–j). Another repeated analysis using the FANTOM5 bulk data further confirmed autocrine edges had higher ranks (Supplementary Fig. 6c–f). Hence, we conclude that autocrine signalling is a major predicted feature of cell-to-cell communication networks.

**Prediction of cellular communities in the Tabula Muris.** To examine whether the summed-specificity weighted cell-connectivity-summary networks might help reveal sets of cell types that work together within an organ or to achieve a biological process, we carried out hierarchical clustering of cell types by the vectors of their summed-specificity weights (Supplementary Fig. 5). For both the secreted ligand and plasma-membrane ligand mediated networks, this failed to reveal any underlying clustering of cell types into organs, tissues or cellular communities. Instead, cells tended to cluster by lineage (indicated as colour bars in Supplementary Fig. 5).

We next examined the top 10 summed-specificity edges based on the secreted and plasma-membrane ligands and visualised them as cell-connectivity-summary networks which revealed distinct cell communities for both secreted and plasma-membrane ligands (Fig. 5). For the connections involving secreted ligands, we observed four disconnected communities (Fig. 5a). The largest community involved hepatocytes broadcasting to basophils, microglial cells, megakaryocyte-erythroid progenitors, and proximal tubule epithelial cells. Proximal tubule

epithelial cells were also predicted to receive signals from pancreatic beta cells and epithelial cells of the trachea. We also predicted two smaller communities of cardiac muscle cells communicating to endocardial cells, and pancreatic delta cells communicating to enteroendocrine cells of the large intestine. Lastly, oligodendrocyte precursor cells (OPC) were predicted to undergo strong autocrine signalling and receive incoming communication from bladder cells. Repeating this analysis using the FANTOM5 bulk primary cell data also predicted a large community with hepatocytes as a central broadcasting node (Supplementary Fig. 6a).

Examining the most specific ligand–receptor pairs involved in each cellular interaction identified both well-known and novel pairs that appear to be biologically relevant (Supplementary Data 6). The most specific ligands driving communication from cardiac muscle cells to endocardial cells included factors relevant to cardiac development and homoeostasis such as *Angpt1*, *Bmp10*, *Vegfa*, *Vegfb*, and *Nppa*. The most specific communication was carried out via *Nppa-Npr3*, where *Npr3* was previously reported as an endocardial marker[26], and *Nppa* was reported to be expressed in cardiomyocytes[27]. These expression patterns have also been recently confirmed in another single-cell analysis of heart[28].

Similarly, for pancreatic delta cells communicating to enteroendocrine cells of the large intestine we also identified somatostatin (*Sst*) as the most specific factor generated by delta cells[29] and confirmed that two of its receptors, *Sstr1* and *Sstr5*, are specifically expressed in enteroendocrine cells[30]. Interestingly, *Sst* has been shown to signal via *Sstr5* to induce mucin production in the large intestine[31]. Additionally, a recent study of intestinal delta cells validated *Sst-Sstr5* mediated signalling to intestinal enteroendocrine cells[32].

In the case of the hepatocyte-centred community, we re-identified the well-known endocrine relationship from hepatocytes to megakaryocyte-erythroid progenitors mediated by *Thpo* and its receptor *Mpl*[33]. In addition to identifying such specific edges, we also predicted that multiple fibrinogens produced by hepatocytes (*Fga*, *Fgb*, and *Fgg*) are used to signal via *Itgb1* (proximal tubule cells), *Itga2b* (megakaryocyte progenitors and

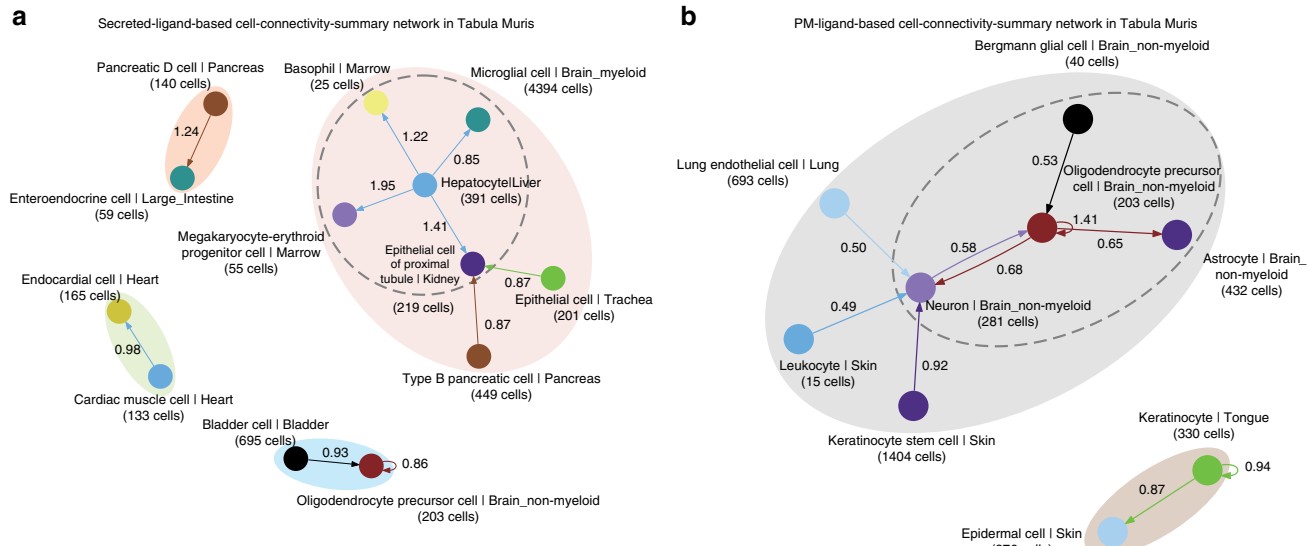

**Fig. 5 Putative secreted and plasma-membrane-ligand mediated communities identified in Tabula Muris.** Visualisation of cell types connected by the top ten summed-specificity edges in Tabula Muris based on **a** secreted ligands and **b** plasma-membrane ligands. Coloured shadows highlight the isolated communities found in each network, edge labels correspond to the summed-specificity weight of the edge. Dashed oval in **a** highlights highly specific signalling from hepatocytes to multiple cell types via secreted ligands. Dashed oval in **b** highlights a set of four brain-derived cell types communicating via plasma-membrane ligands.

basophil), *Itgb2* (basophils) and *Itgam* (basophils and microglial cells).

For the plasma-membrane ligand-mediated interactions, we observed two communities, which consisted of cell types that were more likely to be physically collocated than those seen from the secreted ligand pairs. One community consisted of seven cell types including four brain-derived cell types (neurons, OPCs, astrocytes, and Bergmann glial cells), while the other consisted of keratinocytes predicted to strongly signal with themselves and epidermal cells (Fig. 5b). Repeating this analysis using the FANTOM5 bulk primary cell data also predicted a community of neural cells (Supplementary Fig. 6b). Examining the most specific ligand–receptor pairs involved in signalling within the epidermal cell community (Supplementary Data 6) identified binding between multiple desmogleins and desmocollins (*Dsg–Dsc*), as well as ephrins and eph receptors (*Efn–Eph*). These proteins are known to be expressed in skin and are important to its biological functioning[34–39]. Similarly, for the edges between the four nervous system cell types (neuron, OPC, astrocyte, and Bergmann glial cell) multiple neuroligins were predicted to signal via multiple neurexins and multiple *Slirtk* ligands via *Ptprs*. Interestingly, although neuroligin–neurexin complexes are known key interactors at neuronal synapses, growing evidence indicates that they are also used for interactions involving astrocytes, oligodendrocytes and OPCs[40,41]. We also observed interacting pairs with more restricted cell tropism, such as *Rtn4* signalling to *Lingo1* (used between neurons and OPCs) and *Cntn6*, *Dll3*, *Cntn1* signalling to *Notch1* (used between OPC and astrocytes).

**Differential network analysis in NATMI.** Lastly, we used NATMI to predict age-related changes in cell communication within the murine mammary gland (mammary glands from 3- and 18-month-old mice profiled in the Tabula Muris Senis[42] were compared). A simple edge count analysis, at a detection rate threshold of 20%, revealed that there were substantially more ligand–receptor edges predicted as active at 3 months than at 18 months (2045 edges were detected at both ages; 1247 edges were detected at 3 months only; and 340 edges were detected at 18 months only, Supplementary Data 7 and 8). Examining differences in the cell-connectivity-summary networks based on the 3- and 18-month-old mammary gland (Fig. 6) revealed specific cell types were driving these age-related differences. In particular, edges involving basal cells (basal cell > basal cell, luminal epithelial cell > basal cell, stromal cell > basal cell and basal cell > stromal cell) were more than twofold higher in the 3-month-old mammary gland than the 18-month sample. Similarly, edges involving B and T lymphocytes (B cell > B cell, B cell > T cell and T cell > T cell) were more than twofold higher in the older mice (Fig. 6).

Furthermore, 266 (78.2%) of the 340 edges only detected in the 18-month-old mammary gland involved signalling to or from T and B cells, while only 141 (11.3%) of the 1247 edges exclusively active at 3 months involved lymphocytes. Conversely, 613 (49.2%) of the 1247 ligand–receptor edges only detected at 3 months involved signalling to or from basal cells while only 52 (15.6%) of the 340 edges exclusively active at 18 months involved basal cells.

Examining the basal cell data in more detail identified 9 receptors (*Gpc1*, *Procr*, *Fzd7*, *Itga5*, *Ldlr*, *Tlr2*, *Lrp6*, *Ephb1*, and *Tfrc*) and 8 ligands (*Tgfa*, *Ngf*, *Col5a2*, *Il11*, *Col4a2*, *Jag1*, *Col18a1* and *Hspg2*) at least twofold down-regulated at 18 months (Supplementary Data 8). Notably, many of these top down-regulated ligands and receptors are known to be important in maintenance of normal mammary basal stem cells and are implicated in basal-like and triple negative breast cancer[43–51].

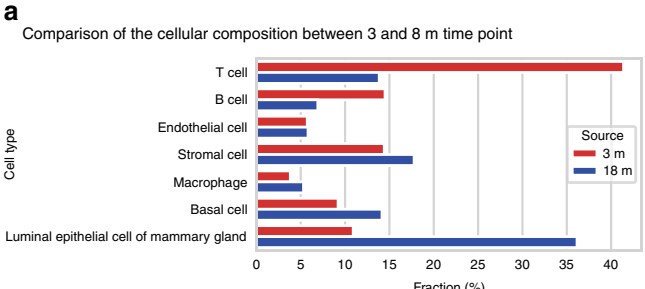

a

Comparison of the cellular composition between 3 and 8 m time point

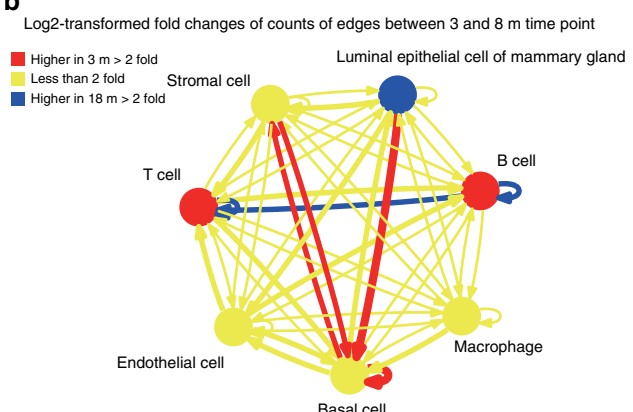

b

Log2-transformed fold changes of counts of edges between 3 and 8 m time point

**Fig. 6 Differential analysis of cell-connectivity-summary networks from aging mammary gland in Tabula Muris. a** Comparison of population fraction of seven cell types in the 3- and 18-month-old murine mammary gland. **b** Comparison of edge-count-based cell-connectivity-summary networks at two time points. Summary edges with twofold or more active ligand–receptor pairs connecting them at 3 months than 18 months are shown in red while those with twofold or more active ligand–receptor pairs connecting them at 18 months than 3 months are shown in blue. Edge thickness is weighted by the log$_2$-transformed fold changes between the two networks. Cell types (nodes) with a twofold difference in cell fraction are shown in red (higher in the 3-month timepoint) and blue (higher in the 18-month timepoint).

## Discussion

Here, we have described NATMI, a tool to help users explore cell-to-cell communication using scRNA-seq or other omics expression data. In the first part of the manuscript, we described the underlying network concepts and explored the effect of different edge weighting approaches. To help identify cell types communicating 'the most', we introduced the concept of cell-connectivity-summary networks and demonstrated how the weight metric and edge filtering used can influence the order of top communicating edges. We recommend using cell specificity weighting for most applications; however, both simple edge count and expression-weighted edges are provided in our outputs as default to yield additional insights on the communicating networks. As a note, the specificity weights were calculated based on the input dataset and thus should be considered local or dataset specific. With larger reference datasets becoming available through efforts such as the Human Cell Atlas[52], we will aim to provide global specificities based on expression across most profiled cell types in future versions of NATMI.

Applying NATMI to the Tabula Muris data (the broadest single-cell atlas to date), we reconfirmed our original findings from FANTOM5 bulk data that autocrine signalling is a major predicted component of cell-to-cell communication networks. This work is also the first to systematically assess the potential for individual cells to express cognate ligand–receptor pairs and to

find substantial potential for their self-signalling. Together, these results predicted that most cell types are major contributors to their own niche[53,54].

In contrast to expectations that cells based in the same tissue would tend to communicate more than those in different tissues, we found essentially no difference between ranks for intra-tissue and inter-tissue edges (Fig. 4c–f) across all cell types within the Tabula Muris atlas. Focusing on the cells connected by the predicted top 10 summary edges identified biologically plausible communities (Fig. 5), which included communities involving cells from the same tissue and communities involving more distant endocrine relationships.

Lastly, we have demonstrated how NATMI can be used to identify changes in cell-to-cell communication by comparing networks predicted from comparable datasets (e.g., paired samples or time-course experiments in which two datasets under consideration have almost identical cellular composition). Using the Tabula Muris Senis dataset, we showed that signalling involving mammary basal cells is down-regulated between 3 and 18 months (Fig. 6). Notably, this is observed both with specificity weighted summary edges and with simple edge counts.

We acknowledge that NATMI is not the first tool to attempt cell-to-cell communication analyses at the single-cell level. Supplementary Data 9 systematically compares features and approaches used in NATMI and 13 other methods[18–21,55–63]. The major discriminating features incorporated in NATMI are that (1) NATMI uses connectomeDB2020, the most comprehensive set of ligand–receptor pairs with primary literature support to date (note, a substantial number of ligand–receptor pairs in other resources lack primary literature support, Supplementary Data 1), (2) NATMI can identify and visualise the cell-types that are communicating the most or the most specifically (both the directed heatmap visualisation and summed-specificity weighting of cell-connectivity summary edges is unique to NATMI) (Fig. 3), (3) NATMI allows users to easily identify and visualise top ligand–receptor pairs based on expression or specificity (Fig. 1) and (4) NATMI allows comparison of networks to identify changes in both ligand–receptor signalling and overall cell-to-cell signalling between cells under different conditions (Fig. 6).

Our comparison to the 13 other tools also highlights three features that are currently not incorporated into NATMI, such as (1) the handling of heteromeric complexes (CellPhoneDB[18], RNA-Magnet[19]), (2) the prediction of downstream signalling consequences (NicheNet[55], SoptSC[59]), and (3) the analysis of spatial transcriptomics data (RNA-magnet[19], SpaOTsc[61]). The modelling of heteromers pioneered in CellPhoneDB[18] acknowledges that many receptors and ligands only function as heteromers. As the full set of biologically relevant heteromers is still yet to be described, modelling heteromers based on what is currently known can result in biases toward well-studied interactors, thus we do not model them in NATMI. Similarly, although using changes in the expression of predicted target genes induced by ligand mediated signalling (used in NicheNet[55]) may provide greater confidence that the ligand–receptor pair is active and inducing a change of state in the target cell we suspect that the predictions are biased toward well-studied ligands and cell types (292 of the ligands in connectomeDB2020 are not in NicheNet). Lastly, although NATMI has not been specifically designed for spatial data, cell type labels based on clusters that incorporate spatial context information (e.g., tumour infiltrating T cells, tumour proximal T cells, tumour distal T cells) can also be analysed.

Furthermore, comparison of NATMI and CellPhoneDB[18] using the same input and ligand–receptor pair lists (Supplementary Notes 1 and 2) revealed NATMI identifies similar edges but is faster, especially with limited computational resources. In Fig. 3, we also showed that edges filtered based on expression or $p$ values (calculated in CellPhoneDB) are very similar whereas the specificity metric used in NATMI identifies a different set of edges, which tend to be more unique and specific to a given pair of communicating cells.

We also acknowledge there are several limitations and caveats to the case studies presented and, more generally, to the methodology used here. First, for the single-cell data analyses we have used the original cellular annotations provided by the authors of each manuscript[11,24,64], which can potentially affect the accuracy of predicted networks. As the quality of user-defined clusters/cell type annotations will define the quality of NATMI predictions, we recommend optimising the clustering/cell type annotation prior to running NATMI. Additionally, the edges that we highlighted in the manuscript are based on experiments without replicates (unfortunately common in current single-cell analyses) making them potentially specific only to the samples tested. Moreover, for single-cell transcriptome data, the predictions are necessarily based on the mRNA expression levels rather than on protein concentration and the data is subject to dropouts where a weakly expressed ligand or receptor may be missed in shallowly profiled cells. As mentioned above, we do not model heterodimerization or requirements for co-receptors. We also do not have estimates of the binding kinetics for each ligand–receptor pair, which can vary across interacting pairs and, ultimately, in conjunction with the number of functional ligand/receptor molecules dictate the physiological response of the interacting pair.

In conclusion, we expect NATMI will greatly facilitate analysis of cell-to-cell communication networks based on single-cell gene expression data as well as for bulk RNA-seq and proteomic data. Note, a distinct advantage of using single-cell data for building these networks is that rare and difficult to isolate cell types can be more easily profiled while avoiding many of the biases introduced by cell isolation/purification and from cell culture needed for bulk profiling. Thus the generation of cell-connectivity-summary networks from this data will allow users to highlight the cell types that are communicating the most and then further explore individual ligand–receptor pairs that are driving these connections. The next exciting, but non-trivial, challenge will be to validate these inferred intercellular communication pathways experimentally (using for example such methods as PIC-seq[65]).

## Methods

**High quality manually curated ligand–receptor interaction database.** To expand upon our previous list of ligand–receptor pairs, we first incorporated pairs from CellPhoneDB v2.0[18]. We downloaded 1144 interactions from https://www.cellphonedb.org/downloads/interactions_cellphonedb.csv and then extracted 693 simple:simple interaction pairs (CellPhoneDB internal ID of format 'CPI-SSxxx'. Heteromeric complexes were excluded). All protein/gene identifiers were updated to HGNC IDs and then compared to our previous catalogue. In total, 478 of these were already in our 2015 resource[1]. For the remaining 215, we used PubMed and google scholar searches to search for primary literature evidence. Finally, 121 pairs with primary literature evidence were kept.

To further expand upon our previous interaction lists we used the following strategy: (1) manual literature search for ligand–receptor pairs using terms 'ligand', 'receptor', 'cytokine', 'growth factor' and for genes from known ligand and receptor families that were not yet covered in the database (2) systematic extraction of PPI pairs from STRINGDB (https://stringdb.org/cgi/download.pl?sessionId=oCkT8UeKh8rN&species_text=Homo+sapiens. Specifically, physical-binding interactions '9606.protein.actions.v11.0.txt' with score ≥700, and experimental interaction '9606.protein.links.full.v11.0.txt' experiments score ≥700) involving known ligands and receptors from the merged list above and putative secreted and plasma-membrane proteins from our previous publication[1]. In total, 3147 putative pairs involving a secreted protein and 1777 involving 2 plasma-membrane proteins were compared to our previous lists and the remainder was manually curated. This combined strategy identified a further 340 ligand–receptor pairs that were absent from our 2015 resource and from CellPhoneDB. During the peer review, we manually reviewed and incorporated an additional 50 pairs from Supplementary Table 3 of RNA-magnet[19], 22 pairs from SingleCellSignalR[20] (source: https://github.com/SCA-IRCM/LRdb/blob/master/LRdb_122019.txt), and 9 pairs from Supplementary Table 1 of ICELLNET[21].

Lastly, feedback from a user of our previous database of one of our entries being incorrect prompted us to re-review the literature support of our original database. This led to exclusion of 143 pairs, mostly from HPRD, where the PubMedID did not actually support the interaction. Excluded putative ligand–receptor pairs from our 2015 resource, CellPhoneDB, RNA-magnet, SingleCellSignalR and ICELLNET are listed along with the reasons for removal in a separate tab of Supplementary Data 1.

**User supplied ligand–receptor interactions (optional)**. Currently, NATMI uses connectomeDB2020 as the default ligand–receptor pair list, but users can also provide their own custom ligand–receptor pairs lists, as described at https://github.com/forrest-lab/NATMI.

**Identification of ortholog ligand/receptors in other vertebrate species**. The curated ligand–receptor database provided in this manuscript is based on interactions between human ligands and receptors. To run NATMI on other species we extracted the homologues of interacting pairs from the NCBI HomoloGene Database[23]. The homologous pairs for 20 commonly requested species including mouse, rat, zebrafish (https://www.ncbi.nlm.nih.gov/homologene/statistics/) can be automatically inferred by NATMI based on the input gene expression data. Note, as there are known species specific ligand–receptor pairs, we recommend to always check the literature and validate reported edges when applying NATMI to other species. For the case studies presented here using mouse single-cell data, we did not systematically review the human–mouse ortholog pairs but we might consider systematically annotating all pairs in a future database release. There are, however, some drawbacks to limiting the pairs to those that have been confirmed in the species of interest only vs a broader survey based on the largest collection of available pairs followed by post analysis confirmation. Our recommendation to end-users is to run NATMI with all L–R pairs available in connectomeDB2020, rank the predicted top L–R pairs and then confirm the pairs of interest in the literature or experimentally. Note it is also highly dependent on the degree of orthology and the coverage/paucity of ligand–receptor studies in the species considered.

**Required input data**. NATMI requires the user to provide a gene expression file (NATMI can process both raw and normalised gene expression data but we recommend users to use a normalisation appropriate for their data), with columns containing expression measurements for a single cell (or bulk measurement) and rows corresponding to genes. For single-cell analyses, the user also needs to provide a metadata file with the mapping between each cell in the dataset and a cell type/cluster label. Step-by-step instructions on how to generate or export these two tables from Seurat[66] and SCANPY[67] are provided online in the GitHub resource: https://github.com/forrest-lab/NATMI. For the expression data table, all datasets used in this paper were normalised by total number of unique molecular identifiers (or reads if the data does not use UMIs) and then rescaled by multiplying by 1,000,000 (i.e., counts per million UMIs/reads).

Using connectomeDB2020 (or a user-specified ligand–receptor pair list) NATMI extracts the expression levels of every ligand and receptor in the provided single-cell (or bulk) dataset. When using single-cell data, NATMI summarises the expression of every ligand and receptor for each cell type/cluster specified in the metadata file (by calculating mean expression, total expression and fraction of cells the gene is detected in). For the analyses presented in this manuscript we consider a gene as expressed in a given cell type if more than 20% of cells of the cell type had non-zero read counts for that gene. The user however is free to alter the detectionThreshold parameter to suit the capture efficiency of their dataset and can use expressionThreshold parameter to filter out weakly expressed ligands and receptors.

**Ligand–receptor edge weights**. NATMI outputs weights of edges from a ligand-producing cell type/cluster to a receptor-expressing cell-type/cluster using three metrics. These are mean-expression weight, specificity weight and total-expression weight. The mean-expression weight is calculated as the product of the mean expression of the ligand in a cell type/cluster and the mean expression of the receptor in a cell type/cluster: $\text{edge(cell-type1} \rightarrow \text{cell-type2)}^{\text{mean}}_{\text{ligand1-receptor1}} = \text{cell-type1}^{\text{mean}}_{\text{ligand1}} \times \text{cell-type2}^{\text{mean}}_{\text{receptor1}}$. The specificity weight is calculated as the product of (1) the mean expression of the ligand in a cell type divided by the sum of the mean expression of the ligand across all cell types in the dataset and (2) the mean expression of the receptor in a cell type divided by the sum of the mean expression of the receptor across all cell types in the dataset: $\text{edge(cell-type1} \rightarrow \text{cell-type2)}^{\text{specificity}}_{\text{ligand1-receptor1}} = \text{cell-type1}^{\text{mean}}_{\text{ligand1}} \times (\Sigma \, (\text{cell-type}^{\text{mean}}_{\text{ligand1}}))^{-1} \times \text{cell-type2}^{\text{mean}}_{\text{receptor1}} \times (\Sigma \, (\text{cell-type}^{\text{mean}}_{\text{receptor1}}))^{-1}$.

Note, we do not use total-expression weight for any of the analyses presented in the manuscript, however it is provided as a feature for future applications. The total-expression weight is calculated as the product of the sum of expression of the ligand in a cell type and the sum of expression of the receptor in a cell-type: $\text{edge (cell-type1} \rightarrow \text{cell-type2)}^{\text{sum}}_{\text{ligand1-receptor1}} = \text{cell-type1}^{\text{sum}}_{\text{ligand1}} \times \text{cell-type2}^{\text{sum}}_{\text{receptor1}}$. Weighting by total-expression acknowledges that each cell type in the network is present at different abundances, thus a cell type that weakly expressed a ligand at a lower mean-expression level than another may still be the major cell type producing the ligand if it is more abundant (e.g., 500 cells of cell type A expressing mean 10CPM of a ligand vs 10 cells of cell type B expressing mean 80CPM of the same ligand → 5000 in cell type A vs 800 in cell type B). Care must be taken in interpreting total-expression weights as they assume unbiased (truly representative) sampling of the cells in the sample.

**Cell-connectivity-summary-network edge weights**. To summarise cell-to-cell connectivity within the network, NATMI generates a matrix of cell-connectivity-summary-network edges. These can be weighted by edge-count or summed expression and specificity. Using the ligand–receptor weights described above users can generate edge-count based summaries that simply count the number of ligand–receptor pairs, from cell-type1 to cell-type2, that pass a set of user-defined thresholds. For example, count all pairs observed at a detectionThreshold of 20%, an expressionThreshold of 10CPM, or with a specificityThreshold of 0.1. In contrast, the summed-specificity weight sums all specificity weights from cell-type1 to cell-type2 without applying a hard threshold. Cell type pairs connected by many specific edges will have high summed-specificity weights.

**Visualising cell-to-cell communication edges using NATMI**. NATMI uses multiple plot types to visualise cell-to-cell communication networks. With the Skelly et al.[11] cardiac dataset as a demonstration, we used NATMI to generate heatmaps of top-ranked ligand–receptor pairs based on expression and specificity weights (Fig. 1) and three different cell-connectivity-summary networks visualisations (heatmap, network graph and circos views, Figs. 2 and 3).

To generate these plots, users can run the following commands: 'python ExtractEdges.py --emFile expression.matrix.txt --annFile annotation.txt --out outfolder' and 'python VisInteractions.py --sourceFolder outfolder'. By default, this will generate nine cell-connectivity-summary-network visualisations (three views with three kinds of edge weighting strategies (summed expression, summed specificity, and edge count)), two heatmaps of top-ranked ligand–receptor pairs based on average expression and specificity weights, and filtered ligand–receptor-mediated edge list, adjacency matrices of cell-connectivity-summary networks.

To visualise delta networks (as shown in Fig. 6), users should first extract edges in the two conditions by running the following commands: 'python ExtractEdges.py --emFile expression.matrix.condition1.txt --annFile annotation.condition1.txt --out condition1.folder' and 'python ExtractEdges.py --emFile expression.matrix.condition2.txt --annFile annotation.condition2.txt --out condition2.folder'. These are then compared using the command: 'python DiffEdges.py --refFolder condition1.folder --targetFolder condition2.folder --out compare.folder'. To visualise the delta networks use the command: 'python VisInteractions.py compare.folder'. Consequently, six heatmaps and six network graphs of the delta networks based on three kinds of edge weighting strategies (averaged expression, summed specificity, and edge count) and two change quantification methods (absolute difference and fold change) are generated. Adjacency matrices of cell-connectivity-summary networks in the two conditions and delta networks are also provided for further exploration.

More details (such as how to apply different edge filters) can be found in NATMI's GitHub tutorial page: https://github.com/forrest-lab/NATMI

**Reporting summary**. Further information on research design is available in the Nature Research Reporting Summary linked to this article.

## Data availability
All data analysed within this paper are publicly available. The Skelly et al.[11] mouse cardiac dataset can be downloaded from ArrayExpress (experiment E-MTAB-6173). The Tabula Muris data can be downloaded from figshare (https://figshare.com/articles/Single-cell_RNA-seq_data_from_Smart-seq2_sequencing_of_FACS_sorted_cells/5715040). The Tabula Muris Senis mammary gland data can also be downloaded from figshare (https://figshare.com/articles/Single-cell_RNA-seq_data_from_microfluidic_emulsion/5715025). The FANTOM5 CAGE expression data for human protein-coding genes in the 144 human primary cells are publicly available on FANTOM5 website (https://fantom.gsc.riken.jp/5/suppl/Ramilowski_et_al_2015/data/ExpressionGenes.txt).

## Code availability
NATMI, an open-source Python tool, and connectomeDB2020 database are available at GitHub (https://github.com/forrest-lab/NATMI). The repository also includes installation instructions, format requirements, detailed function descriptions, example workflows for the user and FAQs section. In addition, we provide the output of an example workflow in the repository so users can preview the analysis results before installing NATMI.

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

## Acknowledgements

We would like to acknowledge and thank Prof. A. Swarbrick, Dr. S. Wu and Dr. R. Tothill for their feedback on NATMI. This work was carried out with the support of a collaborative cancer research grant provided by the Cancer Research Trust 'Enabling advanced single-cell cancer genomics in Western Australia', a grant from Cancer Council of Western Australia and an Australian National Health and Medical Research Council project grant APP1146323. R.H. is supported by an Australian Government Research Training Program (RTP) Scholarship. A.R.R.F. was supported by funds raised by the MACA Ride to Conquer Cancer and a Senior Cancer Research Fellowship from the Cancer Research Trust. A.R.R.F. is currently supported by an Australian National Health and Medical Research Council Fellowship APP1154524. Analysis was made possible with computational resources provided by the Pawsey Supercomputing Centre with funding from the Australian Government and the Government of Western Australia. J.A.R. is supported by a Research Grant from MEXT to the RIKEN Center for Integrative Medical Sciences.

## Author contributions

NATMI was developed by R.H. with help from J.A.R. The project was conceived by A.R.R.F. The updated ligand–receptor pair lists were curated by A.R.R.F., H.T.O., and J.A.R. The paper was written by R.H. and A.R.R.F. with help from E.D. and J.A.R.

## Competing interests

The authors declare no competing interests.

## Additional information

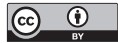

