## [Peer Review File · Nature Communications]

REVIEWER COMMENTS

Reviewer #1 (Remarks to the Author):

Hou et al. present NATMI, a database-based tool to study cell-cell communication through ligand-receptor interactions, using a single-cell RNA sequencing dataset as input. The tools presented are useful, and have potential - through their careful use - to facilitate future biological discovery. I have several questions or concerns about the implementation of NATMI, which I think are important to be addressed for the methods to be clear and for the tools to be useful to the community.

Major points

1. In Methods, on edge weights, according to the authors the specificity-based weight is calculated by taking: "the product of the ligand specificity and the receptor specificity of the interacting cell types" - reliability of this measure relies on the accuracy of the cell clustering in the dataset. For many single-cell datasets, clustering is hard, there are subjective choices to be made, and cell clusters will not always contain well-defined populations of cells (i.e. variability within a cluster can be large). This might put into question the usefulness of specificity-based edge weights. Can NATMI handle such variability? Also: does the use of the specificity measure bias towards self-communication edges? This might affect the autocrine signaling results reported below.
2. "Usually, if 20% of cells in a particular cell type had non-zero read counts for a gene, that gene is defined as "expressed" " - this is important and needs further justification - gene will be recorded as expressed only if very few reads are counted in 20% of cells - is this enough? The method makes no discrimination between this (low) level of expression and a gene expressed in 20% of cells but with hundreds/thousands of counts per cell - is this not an important difference? How sensitive/robust is NATMI to changes in these parameters?
3. Ligands and receptors are not created equal. Expression of receptors may be ubiquitous or widespread without any communication between cells, whereas expression of the ligand is more likely to be correlated with signaling pathway activity (conditional on expression of the receptor). This is a serious limitation to NATMI becoming widely adopted. One way to deal with this is to incorporate target gene expression into edge weights, since such expression can add support to receptor expression in receiver cells. Two current methods for cell-cell communication network inference include target gene expression in their scoring (PMID:30923815, PMID:31819264): only one is cited and even for this, no meaningful discussion of the relevance of target genes is given. Both should at least be discussed, and ideally some such incorporation of the concepts presented in these alternative methods should be performed to mitigate reliance only on receptor expression.
4. On the Tabula Muris data analysis, the statement: "We report that on average each cell type in the Tabula Muris expresses 100 ligands and 128 receptors" (Fig 3) - is this useful? I.e. how much interpretation can really be made from "average number of ligands" summing over whole families, and without any knowledge of their expression levels (see point 2 above). Further: "Within the Tabula Muris data we observed substantial potential for self-signalling with 620/780 (for 10CPM- and 0CPM- thresholds respectively) ligand-receptor pairs" this is missing something? Out of what total # of pairs? "Hence, we conclude that autocrine signalling is a major feature of cell-to-cell communication networks" - see discussion of target genes above (point 3): this statement is too strong. Without target gene expression analysis, one can - at most - say that the cell types analyzed show expression profiles permissive of autocrine signaling. "when applying NATMI to other species, please always check the literature to verify if a reported edge is only seen in humans" - was this done for the Tabula Muris analysis?

5. "taking advantage of using single-cell data we were able to discriminate whether a ligand and its cognate receptor were actually expressed in the same cell or two different cells of the same cell-type" - quantify this beyond "substantial potential" - not just how many L/R pairs, but how many cells per cell type shared how many pairs? A figure would be helpful (the relevant SI Table does not contain any more information; indeed, why is this a table and not a paragraph in the methods?) Beyond the sentence quoted, almost no mention of the specific advantages and challenges of working with single-cell data in the context of the NATMI results is given. How do the networks predicted differ than when using bulk data? What insights are offered? Are any particular strategies implemented to handle the particular sources of noise in these data?

6. A web interface to present at least the connectomeDB2020 database, if not the communication network tools, would greatly benefit adoption of these tools. (the online viz tool for Ramiłowski et al. 2015, was excellent)

Minor points

Fig 1A - I do not see the point/relevance of this cartoon. 1B conveys the same more clearly, Please replace with more informative cartoon or remove.

- Fig 2 not clear what color represents

- Code issues. NATMI was tested on Python 2 with pandas 0.24.2 - these are now outdated and unsupported. For widespread community use, these should really be updated. E.g. I had to replace ".ix" with ".loc" to fix bug of pandas labels (see report below, running on MacOS 10.15.4, Python 3.7.7 and pandas 1.0.3.

Traceback (most recent call last):

```
File "ExtractEdges.py", line 558, in <module>
main(species, opt.emFile, opt.annFile, idType, interDB, interSpecies, opt.coreNum, opt.out)
File "ExtractEdges.py", line 373, in main
em = em.ix[em.max(axis=1)>0,]
File "~/anaconda3/lib/python3.7/site-packages/pandas/core/generic.py", line 5274, in __getattr__
return object.__getattr__(self, name)
AttributeError: 'DataFrame' object has no attribute 'ix'
```

Reviewer #2 (Remarks to the Author):

The authors present a novel computational tool called NATMI that facilitates the prediction of cell-cell communication based on single-cell RNA sequencing data. First, the authors explain the rationale behind the developed method, and then they illustrate the method of a mouse data set Tabula Muris.

The article starts with a reference to the authors own work from 2015 (connectomeDB2015), which develops into a disclaimer that the present manuscript presents an update to their ligand-receptor data base now called connectomeDB2020. While it is good to see continuous improvement of tools and methods, the advancement appears incremental.

Then, the NATMI method is introduced and some previous work in this field has been cited. However, I find that the authors have to a poor job in researching and referencing previous work. In the light of this, their work method development seems incomplete and lacks taking advantage

of previous work. In fact, I do not understand what the novelty of NATMI is at all, and how it compares to previously published work. Also, the explanation of the computational method itself is insufficient. The lack of detail may indicate a lack of sophistication of the method. For instance, it is unclear how the method deals with zero-dropout that is caused by the single-cell nature of the underlying data and how it affects statistics and robustness of the results. In their "Methods" description, the authors mention that a gene is either classified as expressed or non-expressed. I find that approach very problematic considering the dropout problem with this kind of data.

Finally, a series of visualizations of the output of NATMI is presented. Considering the figure, I find the representations not very innovative and rather confusing. I don't see how this is able to help extract useful information. The application to Tabula Muris data set is not helping to convince either. For instance, Fig. 2 a,b,c,d is rather a vast data dump and not very helpful. More care has been spent on Fig 5 which looks more sophisticated but also highly manually curated. Therefore, I don't know if this is a general benefit of the method or just a more polished results figure.

Point by point response to the reviewers comments.

We thank the reviewers for their comments. These have helped us improve the manuscript and NATMI greatly.

Particularly, the comments that several of the figures were extensive data dumps helped us rethink how to best visualize the data and consequently 5 of the 6 main figures have been replaced or reworked. These new visualization options for individual and delta network analyses have been added to NATMI.

As requested we have expanded our comparison to other tools. In the original manuscript we mention CellphoneDB and Nichenet and in a **Supplementary note** showed that when NATMI and CellphoneDB were run using the same input data and ligand-receptor database they yielded comparable results. We now include **Supplementary Table 9** which compares the features of NATMI with 13 other tools aimed at examining cell-to-cell communication analysis. Key distinguishing features include the use of ConnectomeDB2020 as the largest manually curated ligand-receptor pair annotated with primary literature to support each interaction, and the novel edge weighting strategies which allow the user to systematically manage every facet of the cell-to-cell communication network analysis by filtering and combining independent ligand-receptor-mediated signaling pathways.

We also took the opportunity to manually review L-R pairs from these 13 resources (most of them are built upon and expanded our 2015 resource), which resulted in the incorporation of 50 additional pairs from RNA-magnet, 22 from SingleCellSignalR and 9 pairs from ICELLNET. A further 25 new pairs that we curated during review were also added. In total connectomeDB2020 now contains 2,293 literature supported pairs (**Supplementary Table 1**). A brief web interface for connectomeDB2020 (<https://forrest-lab.github.io/NATMI/>) has been set up for biological interpretation and literature searching.

REVIEWER COMMENTS

Reviewer #1 (Remarks to the Author):

Hou et al. present NATMI, a database-based tool to study cell-cell communication through ligand-receptor interactions, using a single-cell RNA sequencing dataset as input. The tools presented are useful, and have potential - through their careful use - to facilitate future biological discovery. I have several questions or concerns about the implementation of NATMI, which I think are important to be addressed for the methods to be clear and for the tools to be useful to the community.

We thank the reviewer for the comments.

Major points

1. In Methods, on edge weights, according to the authors the specificity-based weight is calculated by taking: “the product of the ligand specificity and the receptor specificity of the interacting cell types” - reliability of this measure relies on the accuracy of the cell clustering in the dataset. For many single-cell datasets, clustering is hard, there are subjective choices to be made, and cell clusters will not always contain well-defined populations of cells (i.e. variability within a cluster can be large). This might put into question the usefulness of specificity-based edge weights. Can NATMI handle such variability?

This is an important point, however it is not peculiar to NATMI or cell-to-cell communication methods in general. For example the quality (and granularity) of clustering directly affect the observed gene expression differences between ‘cell types’ and between cell clusters with the same annotation between two different experiments. The specificity metric per se is not the issue. Note, NATMI also calculates edge weights based on simple edge count and expression weighting, however these metrics do not ameliorate poor clustering/cell annotation by the user.

In the discussion of the original submission we included the caveat:

“Firstly, we have used the original cellular annotations provided by the authors of each manuscript, which can potentially affect the accuracy of predicted networks.”

In our revised version we add to this and explicitly state. “As the quality of user-defined clusters/cell-type annotations will define the quality of NATMI predictions, we recommend optimising the clustering/cell-type annotation prior to running NATMI.”

Also: does the use of the specificity measure bias towards self-communication edges? This might affect the autocrine signaling results reported below.

Regarding the results on the autocrine edges, the results are not dependent on the specificity weighting. **Figure 4a** simply counts the fractions of ligands with cognate receptors detected in the same cell type and plots this versus the counts of receptors with cognate ligands detected in

the same cell type (i.e. it is a simple present/absent call). **Figure 4c-f** is based on specificity weights however the observation is robust to edge weight metric as **Supplementary figure 4** shows the same trend using simple (detected/not detected) edge counts.

2. “Usually, if 20% of cells in a particular cell type had non-zero read counts for a gene, that gene is defined as “expressed” ” - this is important and needs further justification - gene will be recorded as expressed only if very few reads are counted in 20% of cells - is this enough? The method makes no discrimination between this (low) level of expression and a gene expressed in 20% of cells but with hundreds/thousands of counts per cell - is this not an important difference? How sensitive/robust is NATMI to changes in these parameters?

We thank the reviewer for this comment.

We used the 20% threshold when carrying out the re-analysis of the mouse cardiac cell to cell communication network (PMID: 29346760) as this is the same detection threshold the authors originally used. Note however that **detectionThreshold** is actually an input parameter in NATMI which can be altered by the user. Similarly, the user can specify an **expressionThreshold** parameter if only highly expressed genes are of interest. Note CellPhoneDB uses a 10% detectionThreshold as default.

We also used a 20% detection threshold for the remaining analyses presented in the manuscript. The argument for this, in addition to the historical use by other groups, is to obtain a balance between dropout failures in the single cell data (failure to detect expression, even though the cell expresses the gene) and false positives due to contamination of a cluster with other cell types. In general, even with the poorest quality clustering, we expect that contamination with another cell type should not be greater than 20%. In terms of detection, Smart-seq2 (considered one of the most sensitive detection methods (Ziegenhain et al. PMID: 28212749)) is estimated to detect transcripts at a sensitivity of approximately 20% (Picelli et al. PMID: 24385147). For other less sensitive scRNA-seq methods the user can set the **detectionThreshold** parameter lower however this needs to be balanced against increasing the possible false positives due to misclustering.

To address the comment we have adjusted the text from:

“Usually, if 20% of cells in a particular cell type had non-zero read counts for a gene, that gene is defined as “expressed”

To

“For the analyses presented in this manuscript we consider a gene as expressed in a given cell type if more than 20% of cells of the cell type had non-zero read counts for that gene. The user however is free to alter the **detectionThreshold** parameter to suit the capture efficiency of their dataset and can use **expressionThreshold** parameter to filter out weakly expressed ligands and receptors.”

3. Ligands and receptors are not created equal. Expression of receptors may be ubiquitous or widespread without any communication between cells, whereas expression of the ligand is more likely to be correlated with signaling pathway activity (conditional on expression of the receptor).

We have shown previously (<https://www.nature.com/articles/ncomms8866/figures/3> - 3c) that there are equal fractions of ubiquitous ligands and ubiquitous receptors. Our recommendation in the current manuscript is to use the product of the specificity [specificity of the ligand x the specificity of the receptor] to capture this. By this metric, pairs involving ubiquitous ligands and ubiquitous receptors are downweighted. Those with one specific partner (ligand or receptor) pairing with a ubiquitous partner have higher weights, and those involving specific ligands and specific receptors have the highest weights. To identify pairs of cells that are communicating via the most specific L-R pairs, we sum these weights across all L-R pairs.

This is a serious limitation to NATMI becoming widely adopted. One way to deal with this is to incorporate target gene expression into edge weights, since such expression can add support to receptor expression in receiver cells. Two current methods for cell-cell communication network inference include target gene expression in their scoring (PMID:30923815, PMID:31819264): only one is cited and even for this, no meaningful discussion of the relevance of target genes is given. Both should at least be discussed, and ideally some such incorporation of the concepts presented in these alternative methods should be performed to mitigate reliance only on receptor expression.

Thank you for this comment.

In the original discussion we mention NicheNet (PMID:31819264), “Major features of other tools that are currently not incorporated into NATMI include the handling of heteromeric complexes (CellPhoneDB), and prediction of downstream signalling consequences (NicheNet)”, however we did not explain why we currently don’t attempt to incorporate this information.

For NicheNet, there are two preconditions.

1. There needs to be a paired experimental design with measurements encompassing the same sets of cell-types under different conditions from which condition-specific differentially expressed ligands and targets can be identified.
2. There needs to be a reliable mapping between a ligand-receptor pair and downstream target genes that will change expression in response to the ligand.

In the case of 1, many (perhaps most) single cell publications to date do not have such paired multi-condition experimental designs. NATMI does not have this restriction. It can be run on single condition experiments or to identify differences in the network in multi-condition experiments (**Figure 6**).

In the case of 2, our greatest concern is that the necessary data on downstream signalling events for most ligands and receptors in the context of each cell type is so incomplete that it would bias the results to well-studied ligand-receptor pairs in well-studied cell models.

In the case of NicheNet a ‘prior knowledge model on ligand–target links’ (<https://www.nature.com/articles/s41592-019-0667-5/figures/1>) is used to link ligands to targets however for a large fraction of ligand-receptor pairs, there is currently no information on the consequences of their downstream signalling. NicheNet provides a ligand-target gene matrix for 688 putative ligands and in the manuscript uses published expression data before/after treatment for 51 ligands to validate the method. Although valuable, these validations are biased towards well-studied ligands. Notably replotting the data from Figure 1 of the NicheNet publication shows that ligands with less than 200 citations have lower AUROC values (see Fig. below). For the remaining 637 ligands in NicheNet there is no information on likely validation rate. Also there are 293 ligands in ConnectomeDB2020 that are not captured in NicheNet (no target matrix provided) and consequently no predictions made. The validation experiments also only show that the method has value in predicting what ligand a particular bulk (mono-)cell culture was treated with. In most single cell datasets the situation is considerably more complex with multiple cell types and hundreds of predicted ligands detected.

Lastly the prior knowledge model is further biased towards interactions recorded in well studied cell lines and easily accessible primary cells.

In conclusion, we have now added **Supplementary table 9** comparing NATMI to multiple other methods including NicheNet (and SoptSC) and extended the text in the discussion to:

“Major features of other tools that are currently not incorporated into NATMI include the handling of heteromeric complexes (CellPhoneDB), and prediction of downstream signalling consequences (NicheNet). We do not yet attempt to model heteromeric complexes suspecting that current knowledge of heteromeric interactors is too incomplete and may exclude pairs where another partner is involved in the heteromer. **Similarly, the approach of predicting key ligands based on changes in downstream targets presented in NicheNet is highly promising for analysis of paired experiments, however, we suspect the predictions to be biased towards well-studied ligands and cell types. Notably, 292 of the ligands in connectomeDB2020 are not in NicheNet.**”

4. On the Tabula Muris data analysis, the statement: “We report that on average each cell type in the Tabula Muris expresses 100 ligands and 128 receptors” (Fig 3) - is this useful? I.e. how much interpretation can really be made from “average number of ligands” summing over whole families, and without any knowledge of their expression levels (see point 2 above).

We thank the reviewer for the comment and partially share the concerns. The main point of this figure, however, was to confirm our original findings using the bulk FANTOM5 data. Specifically that tens to over a hundred ligands and receptors are expressed in each cell type and that there is substantial potential for autocrine signalling.

We have edited this section thusly.

The following paragraph details the motivation for the analysis:

“One of the ultimate aims of developing intercellular communication network methods is to understand the general principles of cell-to-cell communication within multicellular organisms. Previously, analysis of the FANTOM5 (bulk expression) dataset [1] revealed that most cell types express tens to over a hundred different ligands and receptors, and that hematopoietic cells tend to express fewer ligands and receptors than cells from other lineages. Importantly, it also predicted a substantial potential for autocrine signalling, with over 50% of the ligands and receptors detected in each cell type having cognate partners expressed in the same cell type.”

And what was observed.

“At a detection rate threshold of 20% (commonly applied to single cell datasets [11, 25]), most cell types in the Tabula Muris dataset expressed over a hundred ligands and receptors, with hematopoietic cell types expressing fewer ligands/receptors than other lineages (Supplementary Fig. 4a). Notably, almost half of the ligands detected in any given cell type in Tabula Muris had cognate receptors (and vice versa) detected in the same cell type further confirming our previous prediction of large potential for autocrine signalling in cell-to-cell communication networks (Fig. 4a).”

Further: “Within the Tabula Muris data we observed substantial potential for self-signalling with 620/780 (for 10CPM- and 0CPM- thresholds respectively) ligand-receptor pairs” this is missing something? Out of what total # of pairs? “Hence, we conclude that autocrine signalling is a major feature of cell-to-cell communication networks” - see discussion of target genes above (point 3): this statement is too strong. Without target gene expression analysis, one can - at most - say that the cell types analyzed show expression profiles permissive of autocrine signaling.

We thank the reviewer for the comment. As query 5 also deals with self-signalling we have responded there.

We have also softened the statement.

“Hence, we conclude that autocrine signalling is a major **predicted** feature of cell-to-cell communication networks.”

“when applying NATMI to other species, please always check the literature to verify if a reported edge is only seen in humans” - was this done for the Tabula Muris analysis?

No, as the Tabula Muris data analysis was a case study for the use of NATMI, we did not systematically review all human-mouse ortholog pairs. In a future release we may systematically annotate all pairs but there is a tradeoff between keeping those pairs that have only been confirmed in the species of interest vs a broader survey based on the largest collection available followed by post analysis confirmation. We include this statement in the methods as a caveat for the user’s consideration. Our recommendation to end-users is to run NATMI using all ConnectomeDB2020 L-R pairs, rank the predicted top L-R pairs and then confirm them in the literature or experimentally. Note it is also highly dependent on the degree of orthology and the coverage/paucity of ligand-receptor studies in the species considered.

5. “taking advantage of using single-cell data we were able to discriminate whether a ligand and its cognate receptor were actually expressed in the same cell or two different cells of the same cell-type” - quantify this beyond “substantial potential” - not just how many L/R pairs, but how many cells per cell type shared how many pairs? A figure would be helpful (the relevant SI Table does not contain any more information; indeed, why is this a table and not a paragraph in the methods?)

We thank the reviewer. We have now expanded on this important point and added a new figure (**Fig.4b**), Supplementary figure (**Supp. Fig. 4b**) and supplementary table listing all L-R pairs co-detected in at least 20% of a cell type (**Supp. Table 3**).

“Across the 117 cell types in Tabula Muris, a median of 54 ligand-receptor pairs were co-detected in at least 20% of the same cells (from each cell type) at an expression threshold of 10 counts per million (CPM, Fig. 4b, Supplementary Table 3 lists the ligand-receptor pairs co-detected in each cell type). This extends on our original observations of substantial autocrine signalling potential using the FANTOM5 data [1] and is the first finding that cognate ligands and receptors are co-expressed in a substantial fraction of single cells suggesting their potential for self-signalling. Moreover, we observed that a substantial fraction of all ligand-receptor pairs (31%, 719/2,293) were co-detected in at least 20% of cells of at least one cell type at an expression threshold of 10CPM (Supplementary Table 4 and Supplementary Fig. 4b).”

The new fig and supp fig are shown below.

Fig. 4b) Self-signaling potential of each cell type in Tabula Muris. The dashed line indicates the median number of ligand-receptor pairs co-detected in more than 20% of cells of each cell-type (at 10CPM threshold).

Self-signalling potential of each cell type of Tabula Muris

Supplementary Fig 4b) The number of ligand-receptor pairs co-detected in at least one cell type (at the expression level of 0CPM and 10CPM) as a function of the proportion of single-cells co-expressing the L-R pair.

Co-expression of L-R pairs in single cells of each cell type of Tabula Muris

Beyond the sentence quoted, almost no mention of the specific advantages and challenges of working with single-cell data in the context of the NATMI results is given. How do the networks predicted differ than when using bulk data? What insights are offered?

One of the distinct advantages of using single-cell data to build cell-to-cell communication networks is the possibility to study the expression of ligands and receptors across hundreds of cell types without introducing biases from cell culture and isolation/purification. Single cell profiling has revolutionised the field and facilitates profiling of both rare and difficult to isolate cell types (two issues which often limit the application of bulk profiling). We have also shown it is possible to discriminate self-signalling potential from autocrine-signalling potential using single cell data (something impossible with bulk data).

Also as NATMI uses a separate metadata file to label single cells with cell-type it allows flexible reanalysis of single cell data to address different questions. For example, different metadata files can be used to analyse the same data at different levels of granularity, comparing cells of the same cell-type in different transcriptional states, stages of the cell-cycle and at different parts of a trajectory. This flexibility is impossible with bulk expression data.

Whether the networks predicted using bulk data and single cell data (for the same set of cell-types) would differ. The answer is an emphatic yes, but interpretation would be confounded by a number of factors. A meaningful direct comparison of networks built using bulk data and single cell data for the same set of cell-types would require splitting a homogenous tissue/sample into two and for the bulk profiling, tissue dissociation followed by isolation of each cell type (identified in the single cell data) in sufficient purity and quantity to carry out RNA-seq.

We would expect that the bulk would likely have a better detection rate than the single cell data, but there would be the trade off of altered expression profiles caused by methods used to isolate each cell type for the bulk profiling and difficulties in isolating rare cell-types.

We have added the following to the concluding paragraph.

"Note a distinct advantage of using single cell data for building these networks is that rare and difficult to isolate cell types can be more easily profiled while avoiding many of the biases introduced by cell isolation/purification and from cell culture needed for bulk profiling."

Are any particular strategies implemented to handle the particular sources of noise in these data?

NATMI provides two parameters to suppress sources of noise in the single-cell data. The **expressionThreshold** and **detectionThreshold** parameters allow users to either exclude weakly expressed genes or exclude genes that are detected in only a small fraction of cells of a given cell type. By default NATMI sets the **detectionThreshold** to 20% and **expressionThreshold** to 0.

6. A web interface to present at least the connectomeDB2020 database, if not the communication network tools, would greatly benefit adoption of these tools. (the online viz tool for Ramilowski et al. 2015, was excellent)

We thank the reviewer for the comment. The results summarised in our original publication (Ramilowski et al. 2015) was a precompute using the FANTOM5 data. The tool allowed users to select/highlight sets of cell types (nodes) and specific edges (LR pairs) however there was no facility to upload user data and run new analyses.

The rapid uptake of single cell profiling has led to large datasets routinely generated with expression matrices much larger than those from FANTOM5. Our aim with NATMI was to generate a tool for analysis of such data but unfortunately it is not currently realistic for us to allow users to upload single cell level data and analyse it (we would like to establish this in the near future but we don't yet have funding for this). Moreover, of the 13 additional tools for cell-to-cell communication analysis that we reviewed in **Supplementary table 9**, only CellPhoneDB reported having a webtool that would permit upload of user data. At the time of testing, however, it was disabled due to high load on the processing server and users were highly encouraged to download the package for large datasets (<https://www.cellphonedb.org/explore-sc-rna-seq>).

[REDACTED]

A third-party screenshot from <https://www.cellphonedb.org/explore-sc-rna-seq> was shown here

In response to the reviewer we have set up a simple web interface for connectomeDB2020 hosted on GitHub (<https://forrest-lab.github.io/NATMI/>). We will routinely update this manually curated ligand-receptor database and its future dedicated website.

Minor points

Fig 1A - I do not see the point/relevance of this cartoon. 1B conveys the same more clearly, Please replace with more informative cartoon or remove.

We still think 1a has value to explain the basic workflow but we have now moved the original Figure 1 to supplementary materials (**Supplementary Figure 1**).

- Fig 2 not clear what color represents

Thank you, we have replaced figure 2 completely as the network-graph view and the colours made it difficult to interpret (see response to reviewer 2).

Fig. 2 summarises the cell-connectivity-summary network for the above cardiac dataset based on simple edge count using three different visualisations - heatmap, network graph, and circos plot. High connectivity of these networks makes the network graph and circos views not easily interpretable due to over-plotting issues. The heatmap view, however, avoids the problem and reconfirms one of the major predictions of the cardiac study [11] that fibroblasts are the most trophic, with edges from fibroblasts to 10 of the 12 cell types dominating the network.”

- Code issues. NATMI was tested on Python 2 with pandas 0.24.2 - these are now outdated and unsupported. For widespread community use, these should really be updated. E.g. I had to replace “.ix” with “.loc” to fix bug of pandas labels (see report below, running on MacOS 10.15.4, Python 3.7.7 and pandas 1.0.3.

Traceback (most recent call last):

File "ExtractEdges.py", line 558, in <module>

main(species, opt.emFile, opt.annFile, idType, interDB, interSpecies, opt.coreNum, opt.out)

File "ExtractEdges.py", line 373, in main

em = em.ix[em.max(axis=1)>0,]

File "~/anaconda3/lib/python3.7/site-packages/pandas/core/generic.py", line 5274, in __getattr__

return object.__getattr__(self, name)

AttributeError: 'DataFrame' object has no attribute 'ix'

We thank the reviewer for the thorough code review. Widespread community use of NATMI is definitely our goal. We have updated the source code and tested the code using python 2.7.17 version with pandas 0.24.2, XlsxWriter 1.2.8, xlrd 1.2.0, seaborn 0.9.0 , igraph 0.7.1, NetworkX 2.2 and PyGraphviz 1.3.1 and python 3.7.6 version with pandas 1.0.3, XlsxWriter 1.2.8, xlrd 1.2.0, seaborn 0.10.1 , igraph 0.7.1, NetworkX 2.4, PyGraphviz 1.5, bokeh 2.0.2 and holoviews 1.13.2. We will keep an eye on the changes of NATMI's dependencies so that NATMI is compatible with most Python environments.

Reviewer #2 (Remarks to the Author):

The authors present a novel computational tool called NATMI that facilitates the prediction of cell-cell communication based on single-cell RNA sequencing data. First, the authors explain the rationale behind the developed method, and then they illustrate the method of a mouse data set Tabula Muris.

The article starts with a reference to the authors own work from 2015 (connectomeDB2015), which develops into a disclaimer that the present manuscript presents an update to their ligand-receptor data base now called connectomeDB2020. While it is good to see continuous improvement of tools and methods, the advancement appears incremental.

We thank the reviewer for the comments, however, to clarify the manuscript is not just an incremental database update but the combination of the NATMI tool and our updated ligand-receptor pair database connectomeDB2020.

Briefly, the NATMI tool allows users for a great flexibility to run their analysis using provided (ConnectomeDB2020) and custom LR pairs and for virtually any vertebrate species for which the user has abundance (mRNA expression, protein intensities, etc.) data. Further, as we show in **Supplementary Note 1**, NATMI is computationally inexpensive and relatively fast as compared to CellPhoneDB (the closest competitor) making it a tool of choice for using it on laptops and PCs.

The database update is substantial and important as it builds upon our highly utilised resource. (See <https://scholar.google.com.au/scholar?cites=10701290016815814853>; has been cited 198 times as of July 17th 2020). Importantly the ligand-receptor pairs in the database have undergone manual curation and consequently a pubmedID is provided as primary evidence that the ligand and receptor do indeed interact. We believe focusing on primary literature supported L-R pairs will reduce wasted effort investigating large numbers of non-physiologically relevant pairs predicted from high throughput PPI screens.

Lastly our previous connectomeDB2015 database has been used as (or incorporated into) the underlying ligand-receptor database in multiple recent cell-to-cell communication tools (e.g. NicheNet[PMID:31819264], SingleCellSignalR[PMID:32196115], SpaOTsc[PMID:32350282], Kumar et al.[PMID:30404002], CellTalker[PMID:31924475]) thus our updated connectomeDB2020 database is highly likely to be used by the research community.

Then, the NATMI method is introduced and some previous work in this field has been cited. However, I find that the authors have to a poor job in researching and referencing previous work. In the light of this, their work method development seems incomplete and lacks taking advantage of previous work. In fact, I do not understand what the novelty of NATMI is at all, and how it compares to previously published work.

We appreciate this feedback. Although we cited CellPhoneDB, scTensor and NicheNet, we should have done more to highlight the features and advantages of NATMI compared to other tools. We have now included a comparison with 13 other tools (10 published, 3 preprints) as **Supplementary table 9**.

From this table the major differentiating factors are:

1. We have the highest quality and largest set of manually curated literature supported ligand-receptor pairs collected to date. PubmedIDs are provided for every interaction (see **Supplementary table1**). Note that in the same supplementary file we also include an extra tab that lists the L-R pairs that were excluded after our manual review from HPRD, CellPhoneDB, NicheNet, ICELLNET and RNA-magnet (due to reasons such as: no primary literature supporting the interaction, interaction does not involve a ligand-receptor pair, incorrect PubmedID).
2. NATMI provides remarkable flexibility to end users. Acknowledging that cell-to-cell communication networks are large with hundreds of potential edges the key to gaining biological insights and testable hypotheses is the ability to meaningfully summarise the networks and identify important edges. NATMI achieves this by helping users identify key L-R pairs and key cell-cell pairs via multiple criteria and filtering options.
3. To identify key L-R pairs in the network NATMI calculates edge weights based on both expression values and specificity values. It also calculates the fraction of cells expressing the L-R pair. Together this allows users to identify the most highly expressed L-R pairs as well as the most specific ones and combine these metrics with detectionThreshold and expressionThreshold parameters to further filter these lists based on the criteria that best fit their analyses.
4. Key cell-cell pairs are easily identified from the heatmap visualization of NATMI (network graph and circos views are also supported) and allow users to easily see the impact of different edge weight and edge filtering strategies on the summary networks generated.
5. The delta network function of NATMI also allows comparison of networks to identify changes in both L-R signalling and cell-cell signalling between multiple conditions.

We have now added the following text to the discussion to highlight these points:

"NATMI provides end users with a great flexibility to identify key ligand-receptor and cell type pairs by a combination of different edge weighting metrics, filters and visualisation methods. In particular, the use of summed edge specificity to weight edges in cell-connectivity-summary-networks and to identify cell types that are communicating the most specifically is a novel approach used in NATMI. Notably, from a careful review of recent literature, connectomeDB2020 is the most comprehensive manually curated set of a total of 2,293 human ligand-receptor pairs to date (pairs identified from non-human species will be included in the future update)."

Also, the explanation of the computational method itself is insufficient. The lack of detail may indicate a lack of sophistication of the method.

We have substantially written the Methods section and now include an in depth description of the three metrics used to weigh ligand-receptor edges, and how these can be combined with filters into cell-connectivity-summary-network edge weights.

The new text is as follows:

“Ligand-receptor edge weights

NATMI outputs weights of edges from a ligand-producing cell type/cluster to a receptor-expressing cell-type/cluster using three metrics. These are **mean-expression weight**, **specificity weight** and **total-expression weight**. The **mean-expression weight** is calculated as the product of the mean-expression of the ligand in a cell type/cluster and the mean-expression of the receptor in a cell type/cluster: $\text{edge}(\text{cell-type1} \rightarrow \text{cell-type2})^{\text{mean}}_{\text{ligand1-receptor1}} = \text{cell-type1}^{\text{mean}}_{\text{ligand1}} \times \text{cell-type2}^{\text{mean}}_{\text{receptor1}}$. The **specificity weight** is calculated as the product of i) the mean expression of the ligand in a cell type divided by the sum of the mean expression of the ligand across all cell types in the dataset and ii) the mean expression of the receptor in a cell type divided by the sum of the mean expression of the receptor across all cell types in the dataset: $\text{edge}(\text{cell-type1} \rightarrow \text{cell-type2})^{\text{specificity}}_{\text{ligand1-receptor1}} = \text{cell-type1}^{\text{mean}}_{\text{ligand1}} / (\text{cell-type}^{\text{mean}}_{\text{ligand1}}) \times \text{cell-type2}^{\text{mean}}_{\text{receptor1}} / (\text{cell-type}^{\text{mean}}_{\text{receptor1}})$.

Note, we do not use **total-expression weight** for any of the analyses presented in the manuscript, however it is provided as a feature for future applications. The **total-expression weight** is calculated as the product of the sum of expression of the ligand in a cell type and the sum of expression of the receptor in a cell-type: $\text{edge}(\text{cell-type1} \rightarrow \text{cell-type2})^{\text{sum}}_{\text{ligand1-receptor1}} = \text{cell-type1}^{\text{sum}}_{\text{ligand1}} \times \text{cell-type2}^{\text{sum}}_{\text{receptor1}}$. Weighting by **total-expression** acknowledges that each cell type in the network is present at different abundancies, thus a cell type that weakly expressed a ligand at a lower mean expression level than another may still be the major cell type producing the ligand if it is more abundant (e.g. 500 cells of cell type A expressing mean 10CPM of a ligand vs 10 cells of cell type B expressing mean 80CPM of the same ligand -> 5000 in cell type A vs 800 in cell type B). Care must be taken in interpreting **total-expression weights** as they assume unbiased (truly representative) sampling of the cells in the sample.

Cell-connectivity-summary-network edge weights

To summarise cell-to-cell connectivity within the network, NATMI generates a matrix of cell-connectivity-summary-network edges. These can be weighted by **edge-count** or **summed expression** and **specificity**. Using the ligand-receptor weights described above users can generate **edge-count** based summaries that simply count the number of ligand-receptor pairs, from cell-type1 to cell-type2, that pass a set of user defined thresholds. For example, count all pairs observed at a **detectionThreshold** of 20%, an **expressionThreshold** of 10CPM, or with a **specificityThreshold** of 0.1. In contrast, the **summed specificity weight** sums all specificity weights from cell-type1 to cell-type2 without applying a hard threshold. Cell type pairs connected by many specific edges will have high **summed specificity weights**.”

For instance, it is unclear how the method deals with zero-dropout that is cause by the single-cell nature of the underlying data and how it affects statistics and robustness of the results. In their “Methods” description, the authors mention that a gene is either classified as expressed or non-expressed. I find that approach very problematic considering the dropout problem with this kind of data.

We agree with the reviewer that the dropouts are a common problem in single-cell data. Therefore, NATMI does not ask that all cells of a cell type must express the gene to consider it

'expressed'. Instead, similar to previous work (Skelly et al. [PMID: 29346760], CellPhoneDB [PMID: 32103204]), NATMI uses detection in a user-defined minimum percentage of cells to determine if a gene is 'expressed' in a given cell-type. CellPhoneDB's default threshold is 10%, while in this paper we set the **detectionThreshold** to 20% (same as Skelly et al.).

Finally, a series of visualizations of the output of NATMI is presented. Considering the figure, I find the representations not very innovative and rather confusing. I don't see how this is able to help extract useful information. The application to Tabula Muris data set is not helping to convince either. For instance, Fig. 2 a,b,c,d is rather a vast data dump and not very helpful. More care has been spent on Fig 5 which looks more sophisticated but also highly manually curated. Therefore, I don't know if this is a general benefit of the method or just a more polished results figure.

We thank the reviewer for these frank comments.

We have substantially revised the remaining figures based on comments from both reviewers and have used colour-blind compatible palettes throughout.

We have generated the following updated figures:

New **Figure 1** demonstrates identification of the top ranked ligand-receptor pairs based on expression weighting or specificity weighting.

New **Figure 2** compares 3 visualizations (heatmap, network-graph and circos) views available in NATMI and highlights that for cell-connectivity-summary-network visualization that the heatmap view clearly communicates relative edge weights while network graph and circos views have problems with overplotting.

New **Figure 3** uses the heatmap visualization to show the impact of different edge filtering and edge weight metrics alters the perceived key cell-cell pairs within the Skelly *et al.* data.

New **Figure 4** combines the autocrine signalling results in the original figure 4 and 5 and also incorporates a new panel that summarises self signalling potential within the Tabula muris dataset.

New **Figure 6** reduced the original 3 panel version to 2 panels.

Note as a further 106 ligand-receptor pairs were found since our last submission all results have been recalculated using an updated ligand-receptor pair list (2,293 vs 2,187 in the original submission).

Regarding the original **Figure 5**, the colours of nodes, layout and highlighted communities were manually adjusted to make the figure visually appealing for the manuscript. The figure has been regenerated using the updated L-R list and consequently several nodes have changed.

REVIEWER COMMENTS

Reviewer #1 (Remarks to the Author):

In this revisions the authors have addressed many of the original concerns raised and the manuscript as especially the figures is much improved. There are, however, a number of outstanding points of serious concern that need to be addressed.

1. Supplementary table 9 is a useful table. There is however insufficient discussions of these comparison in the main text. A summary the key features from this table that are shared/distinct between NATMI and previous methods is important to explain/motivate the methods developed in the current work. E.g. how different methods define ligand-receptors sensitivity or how they define edge weights. Also some references missing- the references to the previous methods included in this comparison should be cited in the paper.

2. "We have shown previously (<https://www.nature.com/articles/ncomms8866/figures/3> - 3c) that there are equal fractions of ubiquitous ligands and ubiquitous receptors"
- The authors missed the point of the original comment, which was to note the differences between ligands and receptors in their 'active signaling:' the presence of ligands in a cell is sufficient for sending signals, while the presence of receptors is not sufficient. Receptors are necessary, but high expression is possible (even common) in cells with no pathway activity. The specificity metric is useful but is addressing a different point (specific vs ubiquitous expressions).

3. The authors note that when pathways within cell types have annotated target gene informations this produces bias towards well studied pathways. This is true, nevertheless the information given by targets is also valuable. SoptSC introduced the concept of using intracellular target gene information to assess signaling interactions. NicheNet extended this framework, and offers a large information resource on pathways connecting LR interactions to targets. The authors describe this resource in the response letter but there it is not incorporated or even discuss it in the manuscript. Given the flexible framework of NATMI that the authors note, why not to have both? Incorporate target information e.g. using NicheNet resources where available, and predict based on ligand-receptors alone if not. This removes the concern of bias of only highly studied pathways.

4. "No, as the Tabula Muris data analysis was a case study for the use of NATMI, we did not systematically review all human-mouse ortholog pairs..."
- while the authors answer this question in their response letter, there has been no changes made in the paper. This leads to a potential for confusion about how the analyses were done. Inevitably cannot have both a large (unbiased) resource and entirely validated. Please amend the methods to make sure they explain clearly what was done in this paper and what is suggested to do for users.

5. The new Fig. 3 is interesting. It seems to show that filtering by CellPhoneDB achieves similar results to specificity based filtering. Is this the case? In the discussion the point: "Although both tools identify similar edges (Fig. 3), we show that each declares specific edges in dissimilar ways, and that NATMI requires less user intervention."
Has not been explained in light of new Fig 3. How is edge specification dissimilar (Fig. 3c and 3d look more similar to each other than 3a)? In what sense less user intervention?

Minor points

"recommend to use summed specificity for most analyses as this captures specific signalling between cell types (Fig. 3e)"
-typo? Fig. 3d?

Point by point response to the reviewers comments.

We thank the reviewer for these final comments.

REVIEWER COMMENTS

Reviewer #1 (Remarks to the Author):

In this revisions the authors have addressed many of the original concerns raised and the manuscript as especially the figures is much improved. There are, however, a number of outstanding points of serious concern that need to be addressed.

1. Supplementary table 9 is a useful table. There is however insufficient discussions of these comparison in the main text. A summary the key features from this table that are shared/distinct between NATMI and previous methods is important to explain/motivate the methods developed in the current work. E.g. how different methods define ligand-receptors sensitivity or how they define edge weights. Also some references missing- the references to the previous methods included in this comparison should be cited in the paper.

As points 1, 2 and 3 all relate to comparison to other methods (2 and 3 are specifically about NicheNet and SoptSC) we address it here.

Specifically, we have rewritten two paragraphs in the discussion (highlighted below):

1. We cite all 13 methods in the main text
2. We list the major discriminating features of NATMI
3. We highlight 3 features not incorporated into NATMI (heteromers, downstream signalling and spatial data)
4. We explicitly state “changes in the expression of predicted target genes induced by ligand mediated signalling (used in NicheNet) may provide greater confidence that the ligand-receptor pair is active and inducing a change of state in the target cell”.

“We acknowledge that NATMI is not the first tool to attempt cell-to-cell communication analyses at the single cell level. **Supplementary Table 9** systematically compares features and approaches used in NATMI and 13 other methods [18-21, 54-62]. The major discriminating features incorporated in NATMI are that i) NATMI uses connectomeDB2020, the most comprehensive set of ligand-receptor pairs with primary literature support to date (note, a substantial number of ligand-receptor pairs in other resources lack primary literature support, **Supplementary Table 1**), ii) NATMI can identify and visualise the cell-types that are communicating the most or the most-specifically (both the directed heatmap visualization and summed specificity weighting of cell-connectivity summary edges is unique to NATMI) (**Fig. 3**), iii) NATMI allows users to easily identify and visualise top ligand-receptor pairs based on expression or specificity (**Fig. 1**) and iv) NATMI allows comparison of networks to identify changes in both ligand-receptor signalling and overall cell-to-cell signalling between cells under different conditions (**Fig. 6**).

Our comparison to the 13 other tools also highlights three features that are currently not incorporated into NATMI, such as i) the handling of heteromeric complexes (CellPhoneDB, RNA-Magnet), ii) the prediction of downstream signalling consequences (NicheNet, SoptSC) and iii) the analysis of spatial transcriptomics data (RNA-magnet, SpaOTsc). The modelling of heteromers pioneered in CellPhoneDB acknowledges that many receptors and ligands only function as heteromers. As the full set of biologically relevant heteromers is still yet to be described, modelling heteromers based on what is currently known can result in biases towards well-studied interactors, thus we do not model them in NATMI. Similarly, although using changes in the expression of predicted target genes induced by ligand mediated signalling (used in NicheNet) may provide greater confidence that the ligand-receptor pair is active and inducing a change of state in the target cell we suspect that the predictions are biased towards well-studied ligands and cell types (292 of the ligands in connectomeDB2020 are not in NicheNet). Lastly, although NATMI has not been specifically designed for spatial data, cell type labels based on clusters that incorporate spatial context information (e.g. tumour infiltrating T-cells, tumour proximal T-cells, tumour distal T-cells) can also be analysed.

Furthermore, comparison of NATMI and CellPhoneDB using the same input and ligand-receptor pair lists (**Supplementary Note 1 & 2**) revealed NATMI identifies similar edges but is faster, especially with limited computational resources. In **Fig. 3** we also showed that edges filtered based on expression or p-values (calculated in CellPhoneDB) are very similar whereas the specificity metric used in NATMI identifies a different set of edges, which tend to be more unique and specific to a given pair of communicating cells."

We think highlighting these examples of features present and absent in NATMI are sufficient. To compare the features of all 13 methods the reader is directed to **Supplementary Table 9**.

2. "We have shown previously (<https://www.nature.com/articles/ncomms8866/figures/3-3c>) that there are equal fractions of ubiquitous ligands and ubiquitous receptors"

- The authors missed the point of the original comment, which was to note the differences between ligands and receptors in their 'active signaling:' the presence of ligands in a cell is sufficient for sending signals, while the presence of receptors is not sufficient. Receptors are necessary, but high expression is possible (even common) in cells with no pathway activity. The specificity metric is useful but is addressing a different point (specific vs ubiquitous expressions).

We thank the reviewer for further clarification. We have responded to this above.

3. The authors note that when pathways within cell types have annotated target gene informations this produces bias towards well studied pathways. This is true, nevertheless the information given by targets is also valuable. SoptSC introduced the concept of using intracellular target gene information to assess signaling interactions. NicheNet extended this framework, and offers a large information resource on pathways connecting LR interactions to targets. The authors describe this resource in the response letter but there it is not incorporated or even discuss it in the manuscript. Given the flexible framework of NATMI that the authors note, why not to have both? Incorporate target information e.g. using NicheNet resources where

available, and predict based on ligand-receptors alone if not. This removes the concern of bias of only highly studied pathways.

As shown above in response to point 1 we have now expanded on this and specifically state:

“Similarly, although using changes in the expression of predicted target genes induced by ligand mediated signalling (used in NicheNet) may provide greater confidence that the ligand-receptor pair is active and inducing a change of state in the target cell we suspect that the predictions are biased towards well-studied ligands and cell types (292 of the ligands in connectomeDB2020 are not in NicheNet).”

We may incorporate this type of functionality in a future version but for now we would recommend that users who want this functionality to run NATMI and NicheNet and compare the resulting edges.

4. “No, as the Tabula Muris data analysis was a case study for the use of NATMI, we did not systematically review all human-mouse ortholog pairs...”

- while the authors answer this question in their response letter, there has been no changes made in the paper. This leads to a potential for confusion about how the analyses were done. Inevitably cannot have both a large (unbiased) resource and entirely validated. Please amend the methods to make sure they explain clearly what was done in this paper and what is suggested to do for users.

We agree and modified the methods text from:

“Note that some of the reported ligand-receptor pairs might be specific to certain species only, thus when applying NATMI to other species, please always check the literature to verify if a reported edge is valid.”

To:

“Note, as there are known species specific ligand-receptor pairs, we recommend to always check the literature and validate reported edges when applying NATMI to other species. For the case studies presented here using mouse single-cell data, we did not systematically review the human-mouse ortholog pairs but we might consider systematically annotating all pairs in a future database release. There are, however, some drawbacks to limiting the pairs to those that have been confirmed in the species of interest only vs a broader survey based on the largest collection of available pairs followed by post analysis confirmation. Our recommendation to end-users is to run NATMI with all L-R pairs available in connectomeDB2020, rank the predicted top L-R pairs and then confirm the pairs of interest in the literature or experimentally. Note it is also highly dependent on the degree of orthology and the coverage/paucity of ligand-receptor studies in the species considered.”

5. The new Fig. 3 is interesting. It seems to show that filtering by CellPhoneDB achieves similar results to specificity based filtering. Is this the case? In the discussion the point:

“Although both tools identify similar edges (Fig. 3), we show that each declares specific edges in dissimilar ways, and that NATMI requires less user intervention.”

Has not been explained in light of new Fig 3. How is edge specification dissimilar (Fig. 3c and 3d look more similar to each other than 3a)? In what sense less user intervention?

We thank the reviewer for this comment. We agree that figure 3 should have been further interpreted.

We have extended the results as following:

“In Fig. 3 we show how NATMI can be used to filter the network based on expression levels or specificities of the ligands and receptors involved.

Filtering by expression weights (**Fig. 3a**) can provide users with a higher confidence that the ligands and receptors are expressed at sufficient levels. For the cardiac dataset we explored, both the filtered by expression and unfiltered network (**Fig. 2**) yielded, however, a similar conclusion that the fibroblasts are the most trophic. In contrast, filtering on specificity weights (**Fig. 3b**) highlights a different set of top cell-to-cell pairs. In particular, autocrine signalling of Schwann cells, endothelial cells and granulocytes, fibroblast and Schwann cell signalling to endothelial cells, and fibroblast, granulocyte and pericyte signalling to granulocytes is highlighted while the broad signalling from fibroblasts seen in the unfiltered and expression filtered networks is diminished. We next compared our results with those obtained by filtering edges based on p-values calculated by CellPhoneDB [18]. The resulting heatmap (**Fig. 3c**) is similar to that observed for the expression filtered network (**Fig. 3a**) suggesting NATMI may better highlight high specificity edges. [Note, the heatmap shown in **Fig. 3c** should not be confused with those generated by CellPhoneDB which are symmetric. NATMI heatmaps are asymmetric and have direction from the ligand expressing cell type to the receptor expression cell type.] Lastly, the network can also be summarised using the summed specificity weights between each cell type pair (**Fig. 3d**). This generates a similar network to that in **Fig. 3b**, without requiring to set an arbitrary threshold on specificity. Noticeably, as each approach generates a different view of the network and highlights different most-communicating cell type pairs (**Fig. 3e**), users need to consider these differences when interpreting their own cell-to-cell communication networks. In NATMI, the user can choose any of its built-in approaches, however, we recommend to use summed specificity for most analyses as this captures specific signalling between cell types (**Fig. 3d**). Different edge filtering methods are further explained in a concept **Supplementary Fig. 3.**”

In the discussion we have:

“Furthermore, comparison of NATMI and CellPhoneDB using the same input and ligand-receptor pair lists (Supplementary Note 1 & 2**) revealed NATMI identifies similar edges but is faster, especially with limited computational resources. In **Fig. 3** we also showed that edges filtered based on expression or p-values (calculated in CellPhoneDB) are very similar whereas the specificity metric used in NATMI identifies a different set of edges, which tend to be more unique and specific to a given pair of communicating cells”**

We have removed ‘requires less user intervention’ as although correct it is a minor point. Specifically the top ligand and receptor pairs shown in **Figure 1** are automatically identified by ranking on their expression (**1a**) or specificity (**1b**) weights. The equivalent figure in

CellPhoneDB appears to require manual intervention to 'select' pairs for visualization (see legend for panel a <https://www.nature.com/articles/s41596-020-0292-x/figures/3>).

Minor points

“recommend to use summed specificity for most analyses as this captures specific signalling between cell types (Fig. 3e)”
-typo? Fig. 3d?

Thank you. We have now corrected this.